# GSK3 inhibition rescues growth and telomere dysfunction in dyskeratosis congenita iPSC-derived type II alveolar epithelial cells

Rafael Jesus Fernandez[1,2,3], Zachary JG Gardner[1,2,3], Katherine J Slovik[4], Derek C Liberti[2], Katrina N Estep[2], Wenli Yang[4,5], Qijun Chen[6], Garrett T Santini[3], Javier V Perez[6], Sarah Root[7], Ranvir Bhatia[3], John W Tobias[8], Apoorva Babu[9,10], Michael P Morley[9,10], David B Frank[10,11], Edward E Morrisey[4,9,10], Christopher J Lengner[4,12]*, F Brad Johnson[4,6,13]*

[1]Medical Scientist Training Program, Perelman School of Medicine, University of Pennsylvania, Philadelphia, United States; [2]Cell and Molecular Biology Graduate Group, Perelman School of Medicine, University of Pennsylvania, Philadelphia, United States; [3]Perelman School of Medicine, University of Pennsylvania, Philadelphia, United States; [4]Institute for Regenerative Medicine, University of Pennsylvania, Philadelphia, United States; [5]Department of Medicine, Perelman School of Medicine, University of Pennsylvania, Philadelphia, United States; [6]Department of Pathology and Laboratory Medicine, Perelman School of Medicine, University of Pennsylvania, Philadelphia, United States; [7]College of Arts and Sciences and Vagelos Scholars Program, University of Pennsylvania, Philadelphia, United States; [8]Penn Genomic Analysis Core, Perelman School of Medicine, University of Pennsylvania, Philadelphia, United States; [9]Penn Cardiovascular Institute, University of Pennsylvania, Philadelphia, United States; [10]Penn-CHOP Lung Biology Institute, University of Pennsylvania, Philadelphia, United States; [11]Division of Pediatric Cardiology, Department of Pediatrics, Children's Hospital of Philadelphia, Philadelphia, United States; [12]Department of Biomedical Sciences, School of Veterinary Medicine, University of Pennsylvania, Philadelphia, United States; [13]Institute on Aging, University of Pennsylvania, Philadelphia, United States

*For correspondence:
lengner@vet.upenn.edu (CJL);
johnsonb@pennmedicine.upenn.
edu (FBradJ)

**Summary** Dyskeratosis congenita (DC) is a rare genetic disorder characterized by deficiencies in telomere maintenance leading to very short telomeres and the premature onset of certain age-related diseases, including pulmonary fibrosis (PF). PF is thought to derive from epithelial failure, particularly that of type II alveolar epithelial (AT2) cells, which are highly dependent on Wnt signaling during development and adult regeneration. We use human induced pluripotent stem cell-derived AT2 (iAT2) cells to model how short telomeres affect AT2 cells. Cultured DC mutant iAT2 cells accumulate shortened, uncapped telomeres and manifest defects in the growth of alveolospheres, hallmarks of senescence, and apparent defects in Wnt signaling. The GSK3 inhibitor, CHIR99021, which mimics the output of canonical Wnt signaling, enhances telomerase activity and rescues the defects. These findings support further investigation of Wnt agonists as potential therapies for DC-related pathologies.

## Editor's evaluation

This study reports a new cell line model for Dyskeratosis congenita. The authors characterized this model by examining its ability to form organoids and its defects in telomeres. Transcriptomics analysis unveiled defects in Wnt signaling, and treatment with a Wnt agonist could rescue these defects and enhance telomerase activity. Overall, the study is well designed and executed, the data presented are clear and convincing and the conclusions should be of great interest to researchers studying Dyskeratosis congenita and telomere biology.

## Introduction

Dyskeratosis congenita (DC) is a rare genetic disorder characterized by bone marrow failure, skin abnormalities, elevated risk of certain cancers, and liver and pulmonary fibrosis (PF). These pathologies are caused by abnormally shortened and uncapped telomeres arising from deficiencies in telomere maintenance, typically due to defects in the action of telomerase. Significant progress has been made in treating the bone marrow failure of DC patients, particularly via transplantation, but PF remains a major life-limiting pathology (*Agarwal, 2018*; *Dietz et al., 2011*).

PF is a subtype of interstitial pneumonia that is chronic and progressive, replacing the normal lace-like alveolar architecture with patchy, hyperproliferative fibrous tissue (*Lederer and Martinez, 2018*). Current therapies are only modestly effective and do not reverse the underlying fibrosis, and lung transplantation is not always an option (*King et al., 2014*; *Lederer and Martinez, 2018*; *Richeldi et al., 2014*; *Valapour et al., 2020*). While much of the early work in PF pathogenesis focused on unraveling the contributions of fibroblasts, genetic studies of families with a predisposition to PF argue that defects in alveolar epithelial cells and telomeres are key drivers of disease (*Alder et al., 2015*; *Armanios et al., 2007*; *Bullard et al., 2005*; *Cogan et al., 2015*; *Haschek and Witschi, 1979*; *Kropski et al., 2015*; *Maitra et al., 2010*; *Thomas et al., 2002*; *Wang et al., 2009*). Recent work in mice shows that dysfunctional type II alveolar epithelial (AT2) cells, the putative stem cells of alveoli (*Barkauskas et al., 2013*), can lead to a progressive chronic fibrotic response similar to that seen in patients (*Nureki et al., 2018*). Furthermore, many of the same genes which when mutated cause DC have also been linked to familial PF (*Alder et al., 2015*; *Armanios et al., 2007*; *Cogan et al., 2015*; *Kropski et al., 2017*). In sporadic PF, both age and short telomeres are risk factors, and these risks are linked because age is associated with telomere shortening in the lung, particularly in AT2 cells (*Alder et al., 2008*; *Everaerts et al., 2018*). Consistent with a role for telomere dysfunction in driving PF, AT2 cells in sporadic PF express hallmarks of senescence and have shorter telomeres in fibrotic regions than those in non-fibrotic regions (*Disayabutr et al., 2016*; *Kropski et al., 2015*; *Snetselaar et al., 2017*). Additionally, two human Mendelian randomization studies argue that short telomeres are a cause of PF (*Duckworth et al., 2020*; *Telomeres Mendelian Randomization Collaboration et al., 2017*). Murine studies also argue that telomere dysfunction and senescence in AT2 cells can drive PF (*Naikawadi et al., 2016*; *Povedano et al., 2015*; *Yao et al., 2019*). Although causality is thus evident, exactly how AT2 cell telomere dysfunction leads to PF is poorly understood.

Previous work in our lab using mouse and human induced pluripotent stem (iPS) cell-derived organoid models of DC intestinal defects uncovered a positive feedback loop by which telomere capping and canonical Wnt signaling support one another under normal conditions to maintain the intestinal stem cell niche (*Woo et al., 2016*; *Yang et al., 2017*). In the setting of telomere dysfunction, this virtuous cycle becomes vicious: the resulting suppression of Wnt signaling interferes with stem cell function directly, and it also amplifies telomere dysfunction by diminishing Wnt-dependent expression of telomere maintenance factors, including the catalytic subunit of telomerase, *TERT*, and several of the telomere-protective shelterins. These studies demonstrated that Wnt pathway agonists (including GSK3 inhibitors, which mimic the inhibition of GSK3 by Wnts) can rescue these defects, raising the possibility that Wnt agonism could be of therapeutic benefit in DC. Given how telomere dysfunction in AT2 cells appears to drive PF, we wondered if Wnt agonism might be of benefit in PF. Wnt signaling is important for lung epithelial cell development (*Frank et al., 2016*; *Goss et al., 2009*; *Li et al., 2002*; *Li et al., 2005*; *Maretto et al., 2003*; *Okubo and Hogan, 2004*; *Ostrin et al., 2018*; *Shu et al., 2005*) and regeneration of the adult lung in response to injury (*Nabhan et al., 2018*; *Zacharias et al., 2018*). On the one hand, there is evidence that Wnts may drive PF; for example, they have been found to be upregulated in patients with PF (*Chilosi et al., 2003*; *Königshoff et al., 2008*; *Königshoff et al.,*

*2009*). On the other hand, when the canonical Wnt transcriptional effector β-catenin is deleted in AT2 cells, mice are sensitized to bleomcyin induced PF (*Tanjore et al., 2013*). Wnt signaling is complex and context dependent (*Wiese et al., 2018*), and the exact spatial, temporal, and cell-type specificity of Wnt signaling in PF remains an area of intense investigation. It is difficult to extrapolate from observational pathologic studies of fully developed PF to the potential functional impact of Wnts at earlier stages of the disease, and the large number of interacting cell types in PF lungs also makes it challenging to identify primary drivers in such studies. Furthermore, differences between mouse and human telomere biology together with the generally lower susceptibility of mice to PF make mouse modeling difficult. We therefore generated AT2 cell organoids by directed differentiation of human iPSCs (iPSC-derived AT2 [iAT2] cells) to explore how telomere dysfunction might impact their function (*Jacob et al., 2017*; *Jacob et al., 2019*).

By comparing iAT2s that are isogenic except for an introduced mutation in the gene most often mutated in DC, X-linked *DKC1*, we show that mutant iAT2 cells become senescent in concert with telomere shortening and uncapping. iAT2 cells with short, uncapped telomeres exhibit gene expression changes consistent with decreased Wnt signaling, and treatment with GSK3 inhibitors, such as CHIR99021, rescues their growth and telomere dysfunction. These findings raise the possibility that Wnt agonists may be of benefit in rescuing the stem cell and telomere defects of AT2 cells associated with PF in DC patients.

## Results

### Engineering a *DKC1* mutation into iPS cells

To model the AT2 cells from DC patients, we engineered a well-characterized, causal DC mutation in *DKC1* (*DKC1* A386T) (*Agarwal et al., 2010*; *Batista et al., 2011*; *Woo et al., 2016*) into the BU3 *NKX2.1::GFP SFTPC::TdTomato* (NGST) human iPS cell line (*Jacob et al., 2017*). We established an isogenic pair of cell lines: an introduced mutant line and a corresponding unedited wild-type (WT) line (*Figure 1—figure supplement 1A-C*). Both iPS lines maintained markers of pluripotency and normal karyotypes after the introduction of the *DKC1* A386T mutation (*Figure 1—figure supplement 1D*).

Previous work (*Agarwal et al., 2010*; *Batista et al., 2011*; *Woo et al., 2016*) established that iPS cells with the *DKC1* A386T mutation exhibit decreased telomerase activity resulting in telomere shortening with passage. We confirmed that telomerase activity was reduced, and telomeres shortened with successive passages, in the BU3 NGST *DKC1* A386T iPS cell line when compared to its WT control (*Figure 1—figure supplement 2A-C*).

### iAT2 cells with short telomeres fail to form alveolospheres and grow in size

We next differentiated these paired iPS cell lines into iAT2 cells using the protocol developed by Jacob et al. (see *Figure 1A* for differentiation strategy, *Figure 1—figure supplement 3A, B* for representative sorting strategies). Using iPSCs 35 passages after the introduction of the mutation yielded iAT2s that initially grew in a similar fashion to WT, but which over time developed a growth defect characterized by lower alveolosphere formation efficiency as well as smaller alveolospheres. The phenotype became apparent by 50 days of culture (D50) and was dramatic by D70 (*Figure 1A–C*). In contrast, using iPSCs only five passages after the introduction of the mutation yielded iAT2s with less dramatic defects in alveolosphere growth and size at D70 (*Figure 1—figure supplement 4A-C*). These data indicate that the defects observed were due to progressive telomere shortening after introducing the *DKC1* mutation, and not the immediate effects of telomerase deficiency (or other potential deficiencies) caused by the *DKC1* A386T mutation per se.

Surfactant protein C (SFTPC) is a highly specific marker of AT2 cells (*Kalina et al., 1992*), and the yield of *SFTPC::TdTomato+* (*SFTPC+*) cells was reduced significantly at D70 in *DKC1* mutant cultures, while the percentage of *SFTPC+* cells generated at each time point was not different, suggesting that there is a defect in AT2 cell proliferation or survival (*Figure 1D*). Sorted *SFTPC+* cells from iAT2 cell alveolospheres maintained expression of multiple AT2 specific genes suggesting that the introduced mutation did not affect lineage specification (*Figure 1E*). Thus, DC iPS cells can generate iAT2 cell alveolospheres; however, these alveolospheres lose the capacity to self-renew with successive passaging.

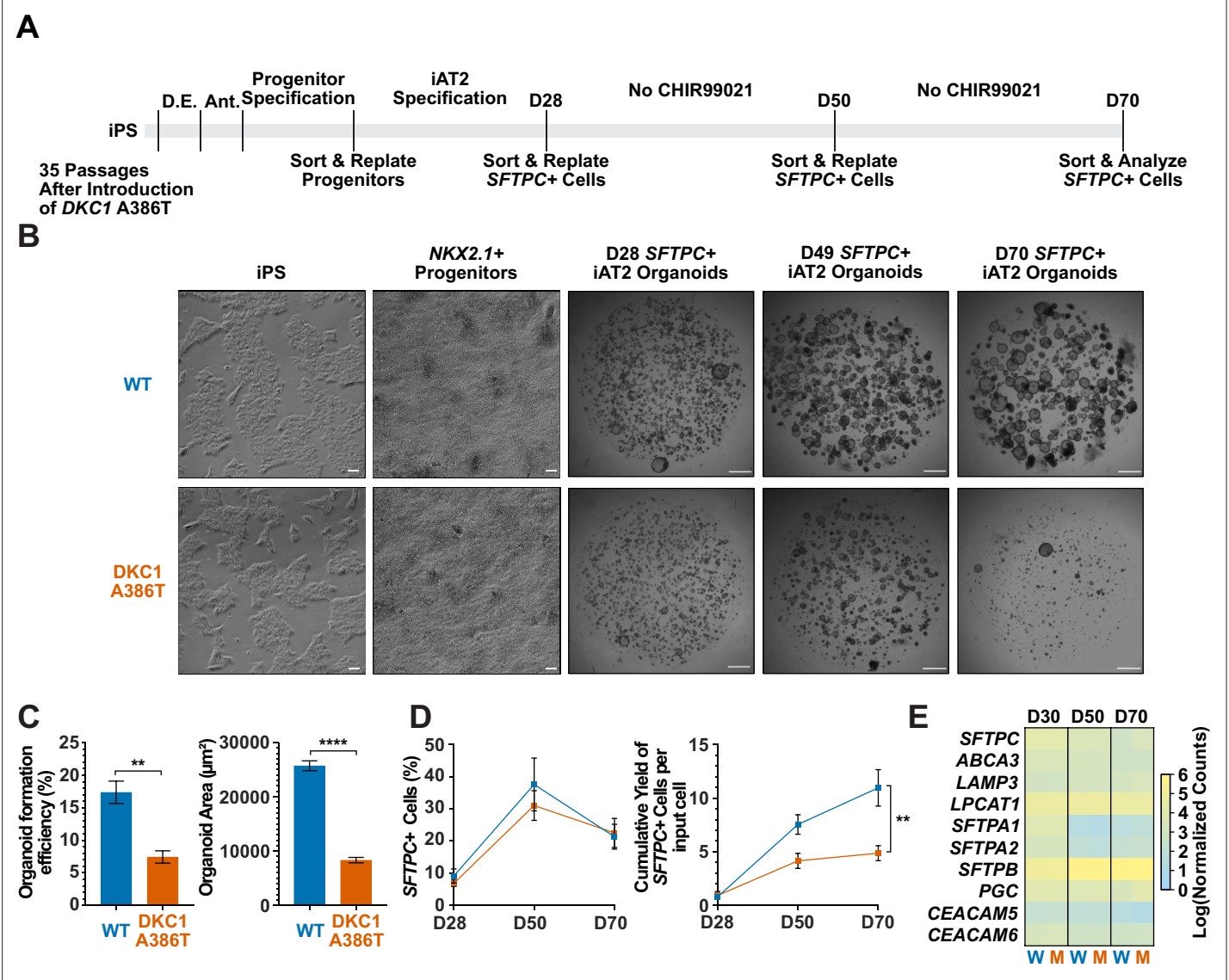

**Figure 1.** Dyskeratosis congenita induced pluripotent stem cell (iPSC)-derived type II alveolar epithelial cells (iAT2s) fail to form alveolospheres with successive passaging. (**A**) Differentiation protocol used to probe the effects of the *DKC1* A386T mutation on iAT2 cells. D.E., definitive endoderm specification; Ant., anteriorization. (**B**) Representative images of differentiating wild type and *DKC1* A386T mutant bearing iAT2 alveolospheres (scale bars, 100 μm as indicated for iPS and *NKX2.1+* progenitors; 1 mm for all alveolosphere images). (**C**) Quantifications of alveolosphere area and formation efficiency on D70 (n = 4; ** p<0.01, **** p<0.0001, and Student's t-test). (**D**) Quantification of the percentage *SFTPC+* cells and the number of *SFTPC+* cells produced with passage of the iAT2 cells shows *DKC1 A386T* iAT2 alveolospheres accumulate fewer *SFTPC+* cells (n = 4; ** p<0.01, and Student's t-test). (**E**) RNA-seq of sorted *SFTPC+* iAT2 cells at different passages shows AT2 cell genes are not grossly affected by the *DKC1* A386T mutation (n = 4, W = wild type, M = mutant).

The online version of this article includes the following source data and figure supplement(s) for figure 1:

**Source data 1.** Source Data for *Figure 1C-E*.

**Figure supplement 1.** Introduction of the *DKC1 A386T* mutation into the BU3 *NKX2.1::GFP SFTPC::TdTomato* (NGST) induced pluripotent stem (iPS) cell line.

**Figure supplement 1—source data 1.** Source data for *Figure 1—figure supplement 1B,C and E*.

**Figure supplement 2.** *DKC1* A386T induced pluripotent stem (iPS) cells show decreased telomerase activity and telomeres shorten with passage.

**Figure supplement 2—source data 1.** Source data for *Figure 1—figure supplement 2A-C*.

**Figure supplement 3.** Representative sorting strategies for *NKX2.1+* progenitors and *SFTPC+* induced pluripotent stem cell-derived type II alveolar epithelial (iAT2) cells.

*Figure 1 continued on next page*

Figure 1 continued

**Figure supplement 4.** Differentiation of induced pluripotent stem cell (iPSC)-derived type II alveolar epithelial cell (iAT2) alveolospheres from early passage iPS cells yields no growth defect.

## DC iAT2 cells derived from iPS cells 35 passages after introduction of the mutation develop hallmarks of senescence at late passage (D70)

To better understand the AT2 cell defects, we further compared the WT and mutant iAT2 alveolospheres at different passages. RNA-seq-based gene expression analyses over successive passages of sorted *SFTPC+* cells from iAT2 alveolospheres showed decreases in proliferation markers (*MKI67* and *MCM2*) as well as an increase in expression of the cell cycle inhibitor *CDKN1A* (p21), and these changes were most pronounced in mutant alveolospheres (*Figure 2A*). Immunofluorescence microscopy based analyses of D70 mutant iAT2 alveolospheres showed an increase in DNA damage marked by 53BP1 foci (*Figure 2B*), and an increased fraction of cells expressing p21 protein (*Figure 2C*), but no increase in apoptosis (*Figure 2—figure supplement 1A-C*). Measuring telomere length using qPCR, DC, and WT iAT2 cells showed no significant change in average telomere length with passage, although average telomere lengths in mutants trended shorter than in WT (*Figure 2—figure supplement 2A*). However, measuring telomere lengths using TeSLA (*Lai et al., 2017*), which is more sensitive for the detection of short telomeres than most other techniques, revealed that DC iAT2 cells had a preponderance of short telomeres at D70 (*Figure 2E–F*). Consistent with this, DC iAT2 alveolospheres showed an increased number of telomere dysfunction induced foci (TIFs), a hallmark of uncapped telomeres (i.e., telomeres that signal DNA damage responses and cell cycle checkpoint arrest *Takai et al., 2003*; *Figure 2D*). These findings indicate that the short and uncapped telomeres that accumulate with passage of DC iAT2 cells lead them to senesce.

## RNA-seq reveals pathways differentially expressed in DC iAT2 cells, including those related to Wnt signaling

To further understand changes in the DC iAT2 cells, we more broadly evaluated RNA-seq-based gene expression changes of sorted *SFTPC+* iAT2 cells. (*Figure 3A*). We found very few significantly differentially expressed genes when comparing WT and mutant cells at D28 and D50, but a large number of differentially expressed genes at D70 (*Figure 3B*), arguing that the gene expression changes seen at D70 are likely driven by uncapped telomeres. Gene set enrichment analyses (GSEA) revealed an upregulation of the DNA damage response, the unfolded protein response (UPR), mitochondrial related functions (oxidative phosphorylation, the respiratory electron chain), a downregulation of hypoxia related signaling, and hedgehog signaling along with other changes (see *Supplementary file 1* for a full list). Ingenuity pathway analysis (IPA) revealed similar changes as well as defects in multiple pathways controlled by inflammatory cytokines like IL1β, IL6, IL17, and others (see *Supplementary file 2* for the full lists). We found a marked upregulation in DC iAT2 cells of many pathways associated with PF (see *Supplementary file 3* for curated list of PF related pathways, see *Supplementary files 1 and 2* for the unedited analyses). These included the UPR (*Lawson et al., 2008*; *Mulugeta et al., 2005*), thyroid hormone metabolism (*Yu et al., 2018*), p53 signaling (*Shetty et al., 2017*), mitochondrial dysfunction and mitophagy (*Chung et al., 2019*; *Yu et al., 2018*), and caveolin function (*Wang et al., 2006*). This analysis also showed an upregulation of non-canonical Wnt signaling (*Figure 3C*), which correlated with a significant upregulation in *WNT5A* and *WNT11*, known non-canonical Wnt ligands (*Figure 3D*). Furthermore, almost every *FZD* gene, encoding co-receptors for canonical Wnt signaling, was downregulated in DC iAT2 cells (*Figure 3E*). Also, GSEA found genes with TCF7 targets in their promoters are downregulated in DC iAT2 cells at D70 (*Figure 3F*). IPA of master regulators at D70 revealed a decrease in genes controlled by lithium chloride (which can potentiate Wnt signaling by inhibiting GSK3) and TCF7 along with an upregulation of genes usually stimulated by Wnt pathway inhibitors (*Figure 3G*). GSEA also revealed a significant downregulation of targets of *miR34a*, a miRNA that we previously demonstrated negatively regulates many components of the Wnt pathway in response to uncapped telomeres (*Figure 3H*; *Yang et al., 2017*). The genes encoding four of the six shelterins, proteins that bind and help maintain normal telomere function, are direct targets of the canonical Wnt transcriptional effector β-catenin, and two of these, *TINF2* and *POT1,* were downregulated in D70 DC iAT2 cells, which may contribute to telomere uncapping

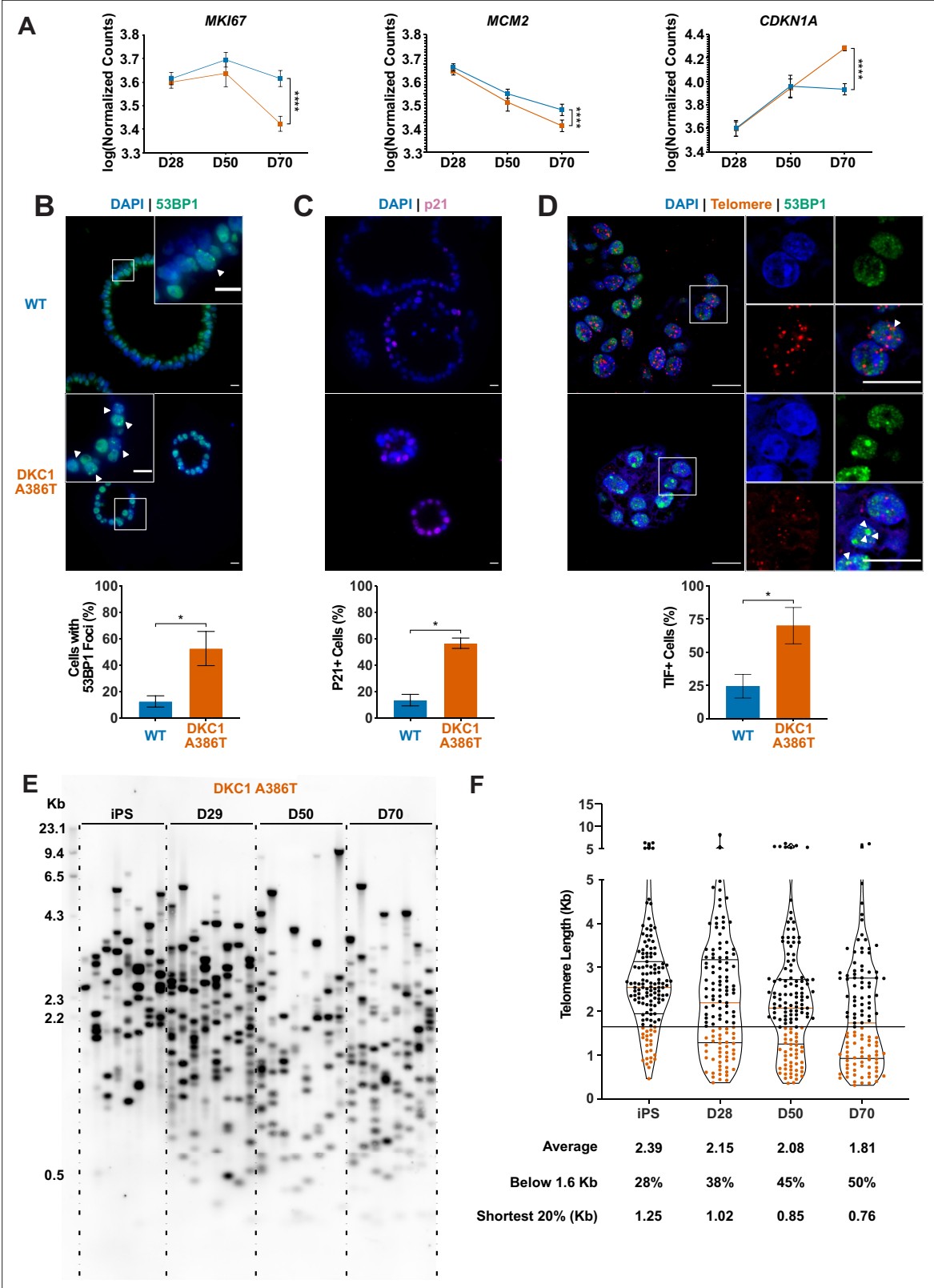

**Figure 2.** Dyskeratosis congenita induced pluripotent stem cell-derived type II alveolar epithelial cells (iAT2s) 35 passages after introduction of the mutation at D70 show hallmarks of senescence. (**A**) Gene expression profiling of iAT2 cells at D28 and D50 shows no difference between wild type and *DKC1* A386T in expression of markers of proliferation and a cell cycle inhibitor, while at D70 cells, there is a significant decrease in *MCM2* and *MKI67* as well as a significant increase in *CDKN1A* (**p21**) (n = 4, **** p<0.0001, DEseq2 pairwise contrast statistics). (**B**) At D70, *DKC1* A386T mutant iAT2 cells

*Figure 2 continued on next page*

*Figure 2 continued*

have a higher fraction of cells with 53BP1 foci (n = 4, * p<0.05, and Student's t-test; scale bars, 10 µm; insets highlight cells with 53BP1 foci as noted by the white arrowheads). (**C**) At D70, *DKC1* A386T mutant iAT2s have a higher fraction of cells positive for p21 (n = 4, * p<0.05, and student's t-test; scale bars, 10 µm). (**D**) At D70, *DKC1* A386T mutant iAT2s have a higher fraction of cells with telomere dysfunction induced foci (TIFs) (n = 4, * p<0.05, and Student's t-test; scale bars, 10 µm; insets highlight cells with TIFs, each one noted by a white arrowhead). (**E**) Representative TeSLA of *DKC1* A386T iAT2 alveolospheres shows telomeres shorten with passage. (**F**) Quantification of *DKC1* A386T iAT2 cell telomere lengths shows a preponderance of short telomeres appears as the iAT2 cells approach D70, red colored data points highlight telomeres under the 1.6 kb threshold (n = 2, 'shortest 20%' reports the 20th percentile of telomere length, in Kb).

The online version of this article includes the following source data and figure supplement(s) for figure 2:

**Source data 1.** Source data for *Figure 2*.

**Figure supplement 1.** Dyskeratosis congenita induced pluripotent stem cell-derived type II alveolar epithelial cells (iAT2s) 35 passages after introduction of the mutation at D70 do not show an increase in apoptosis.

**Figure supplement 1—source data 1.** Source data for *Figure 2—figure supplement 1C*.

**Figure supplement 2.** Telomere qPCR reveals a trend toward shorter average telomere lengths in dyskeratosis congenita (DC) induced pluripotent stem cell-derived type II alveolar epithelial (iAT2) cells.

**Figure supplement 2—source data 1.** Source data for *Figure 2—figure supplement 2*.

beyond simple telomere shortening (*Figure 3I*; *Yang et al., 2017*). These genes were confirmed to be downregulated by ddPCR, and furthermore, DC iAT2 cells expressed lower levels of both *TERT* and *TERC* (*Figure 3—figure supplement 1*). These data indicate that Wnt signaling in AT2 cells is greatly affected by shortened telomeres and that non-canonical Wnt signaling may be elevated while canonical (β-catenin-dependent) signaling may be diminished.

## GSK3 inhibitors rescue the growth of DC iAT2 alveolospheres

Given our previous work showing that GSK3 inhibition reverses telomere dysfunction and associated defects in intestinal models of DC (*Woo et al., 2016*; *Yang et al., 2017*), we attempted to rescue the DC iAT2 alveolosphere formation defect by treatment with CHIR99021, a well-characterized GSK3 inhibitor, which thus stabilizes β-catenin and upregulates canonical Wnt target genes (*Figure 4A*). CHIR99021 partially rescued the alveolosphere growth defect of iAT2 cells in a dose dependent fashion (*Figure 4B–C*). Furthermore, if alveolospheres were cultured continuously prior to D70 with CHIR99021, this prevented the growth defect from emerging (*Figure 4—figure supplement 1A-B*). We also tested another GSK3 inhibitor, CHIR98014, which similarly rescued growth of the mutant iAT2 cells (*Figure 4—figure supplement 2A-B*). These findings suggest that GSK3 inhibition rescued the growth of DC iAT2 alveolospheres.

## CHIR99021 downregulates senescence markers and resolves TIFs in DC iAT2 alveolospheres

We ultimately investigated how CHIR99021 affects the telomere status of DC iAT2s. CHIR treated iAT2 cells showed fewer cells with 53BP1 foci, fewer p21+ cells, and fewer TIF+ cells (*Figure 4D–F*). TeSLA revealed a modest increase in average telomere length but no apparent decrease in the frequency of shortest detectable telomeres (*Figure 4—figure supplement 3A-C*). iAT2 cell telomerase activity was restored in mutant cells treated with CHIR99021 to levels similar to those seen in WT cells (*Figure 4G*). Together, these data argue that GSK3 inhibition not only rescues the growth of DC iAT2 cells, but it also rescues telomere defects, most likely through upregulation of telomerase activity that could extend telomeres that are shorter than those that can be detected by TeSLA; it is also possible that telomerase is contributing to telomere capping via lengthening-independent mechanisms (*Perera et al., 2019*). Of note, withdrawal of CHIR99021 during only 14 days prior to D70 was sufficient to elicit the growth defect (*Figure 4—figure supplement 1A-B*), consistent with an only modest increase in telomere length associated with the increased telomerase activity.

## Discussion

We used isogenic human iPS cell lines to generate DC mutant iAT2 cells with shortened telomeres to interrogate how telomere dysfunction can affect AT2 cell function. We found that shortened and

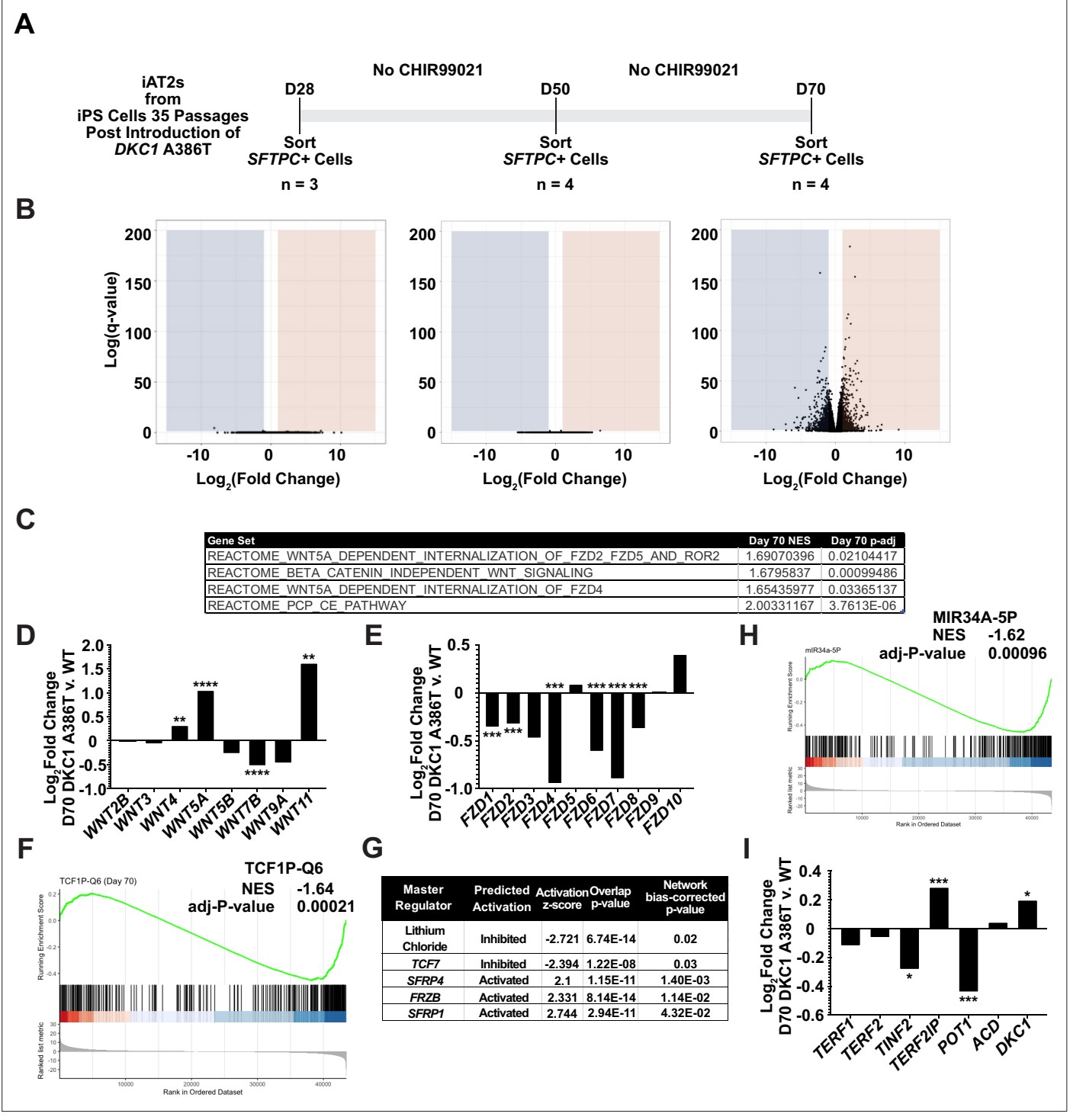

**Figure 3.** RNA-seq of passaged induced pluripotent stem cell-derived type II alveolar epithelial (iAT2) cells reveals a time dependent change in Wnt signaling. (**A**) A schematic to show how cells were prepared for RNA-seq. (**B**) Volcano plots at D28, D50, and D70 show how the number of differentially expressed genes increases at D70. (**C**) Gene set enrichment analyses (GSEA) at D70 comparing *DKC1* A386T iAT2 cells reveal an upregulation of non-canonical Wnt signaling and the planar cell polarity (PCP) pathway. (**D**) RNA-seq analysis shows upregulation of *WNT5A* and *WNT11*, non-canonical WNTs associated with pulmonary fibrosis (* p<0.05, ** p<0.01, *** p<0.001, **** p<0.0001, DEseq2 pairwise contrast statistics). (**E**) RNA-seq analysis shows broad downregulation of many *FZD* receptors (* p<0.05, ** p<0.01, *** p<0.001, **** p<0.0001, DEseq2 pairwise contrast statistics). (**F**) GSEA at D70 comparing *DKC1* A386T iAT2 cells reveals a downregulation of genes with *TCF7* bindings sites in their promoters. (**G**) Ingenuity pathway analysis reveals master regulators at D70 including downregulation of regulation associated with lithium chloride and *TCF7*, and upregulation of regulation

*Figure 3 continued on next page*

*Figure 3 continued*

associated with multiple Wnt inhibitors. (**H**) GSEA at D70 comparing *DKC1* A386T iAT2 cells reveals a downregulation of genes with *miR34A* binding sites. *TINF2* and *POT1* are downregulated in *DKC1* A386T iAT2 cells at D70 (* p<0.05, ** p<0.01).

The online version of this article includes the following source data and figure supplement(s) for figure 3:

**Figure supplement 1.** ddPCR demonstrates key shelterin and telomerase related genes are downregulated in dyskeratosis congenita (DC) induced pluripotent stem cell-derived type II alveolar epithelial (iAT2) cells.

**Figure supplement 1—source data 1.** Source data for *Figure 3—figure supplement 1*.

uncapped telomeres are associated with a defect in alveolosphere formation by iAT2 cells. This defect is characterized by senescent iAT2 cells that upregulate many pathways associated with PF including the UPR, mitochondrial biogenesis and function, thyroid hormone signaling, and p53 signaling. DC mutant iAT2 cells also suppress canonical Wnt signaling, and consistent with this, GSK3 inhibition rescues telomerase activity, telomere capping, and alveolosphere formation. This system provides a new preclinical model to better understand PF pathogenesis and how potential new PF therapeutics affect AT2 cell function in the context of telomere dysfunction.

Wnt signaling is complex and can be broken down into two major categories: β-catenin dependent (canonical) signaling and β-catenin independent (non-canonical) signaling. These distinctions can also be blurred as evidenced by studies that show how non-canonical ligands, such as *WNT5A*, can activate both arms of Wnt signaling (*van Amerongen et al., 2012*; *Mikels and Nusse, 2006*). These complexities therefore make the conflicting reports about whether β-catenin dependent Wnt signaling is of benefit (*Tanjore et al., 2013*) or of harm in PF (*Douglas et al., 2006*; *Henderson et al., 2010*; *Kim et al., 2011*; *Königshoff et al., 2008*; *McDonough et al., 2019*) unsurprising. Given the heterogeneity of the disease both in space and time and the context dependence of Wnt signaling, these studies can often only capture a snapshot of the fibrotic response. Furthermore, there are clear differences between mice and humans with regard to telomere and lung biology (*Basil and Morrisey, 2020*; *Gomes et al., 2011*). These limitations highlight the need for human models capable of assessing the spatial, temporal, and cell type specific properties of Wnt signaling in PF pathogenesis.

Our study also provides evidence of upregulation of β-catenin-independent signaling in DC iAT2 cells and that re-activating β-catenin dependent Wnt signaling using GSK3 inhibitors might provide support for AT2 cell proliferation in the context of telomere dysfunction. It is tempting to extrapolate from other models of lung disease to understand how the activity of the β-catenin dependent Wnt pathway might be of benefit in PF. β-catenin dependent Wnt signaling improved regeneration and survival in a model of emphysema (*Kneidinger et al., 2011*), and inhibition of *WNT5A*, and thus presumably some component of β-catenin independent Wnt signaling, improved repair in a model of COPD (*Baarsma et al., 2017*). Our work, consistent with previous studies, argues that β-catenin dependent Wnt signaling supports AT2 cell telomere capping and proliferation, which may be of benefit during regeneration and repair (*Nabhan et al., 2018*; *Uhl et al., 2015*; *Zacharias et al., 2018*).

Our previous work uncovered a positive feedback loop between Wnt signaling and telomeres in the intestine (*Woo et al., 2016*; *Yang et al., 2017*). Here, we show that aspects of the Wnt-telomere feedback loop appear to be at play in AT2 cells (*Fernandez and Johnson, 2018*), arguing that this connection between Wnt and telomeres is present not just in proliferative tissues such as the intestine, but also in lung cells, which are normally quiescent but proliferate in response to injury. Previous work highlighted the importance of telomerase during alveolar regeneration (*Driscoll et al., 2000*; *Lee et al., 2009*). Furthermore, given previous demonstrations that Wnt can stimulate, directly or indirectly, *TERT* and *TERC* expression as well as telomerase activity in several cell types (*Baena-Del Valle et al., 2018*; *Hoffmeyer et al., 2012*; *Jaitner et al., 2012*; *Zhang et al., 2012*), we anticipated and were able to demonstrate (i) that *TERT* and *TERC* transcripts are reduced in senescent iAT2 cells and (ii) that GSK3 inhibition can restore telomerase activity in this context. Of note, CHIR99021 is known to induce proliferation in a wide variety of cell types (*Hesselbarth et al., 2021*; *Jacob et al., 2017*), which likely contributes to the growth stimulation observed in both WT and *DKC1* mutant iAT2s shown in *Figure 4—figure supplement 1*. This raises the possibility that GSK3 inhibition has additional effects independent of telomeres. We also note that the effects of CHIR99021 could differ in some ways from canonical Wnt signaling, by impacting targets other than GSK3 or by differentially affecting the numerous pathways that are downstream of GSK3. However, the Wnt-related transcriptional changes

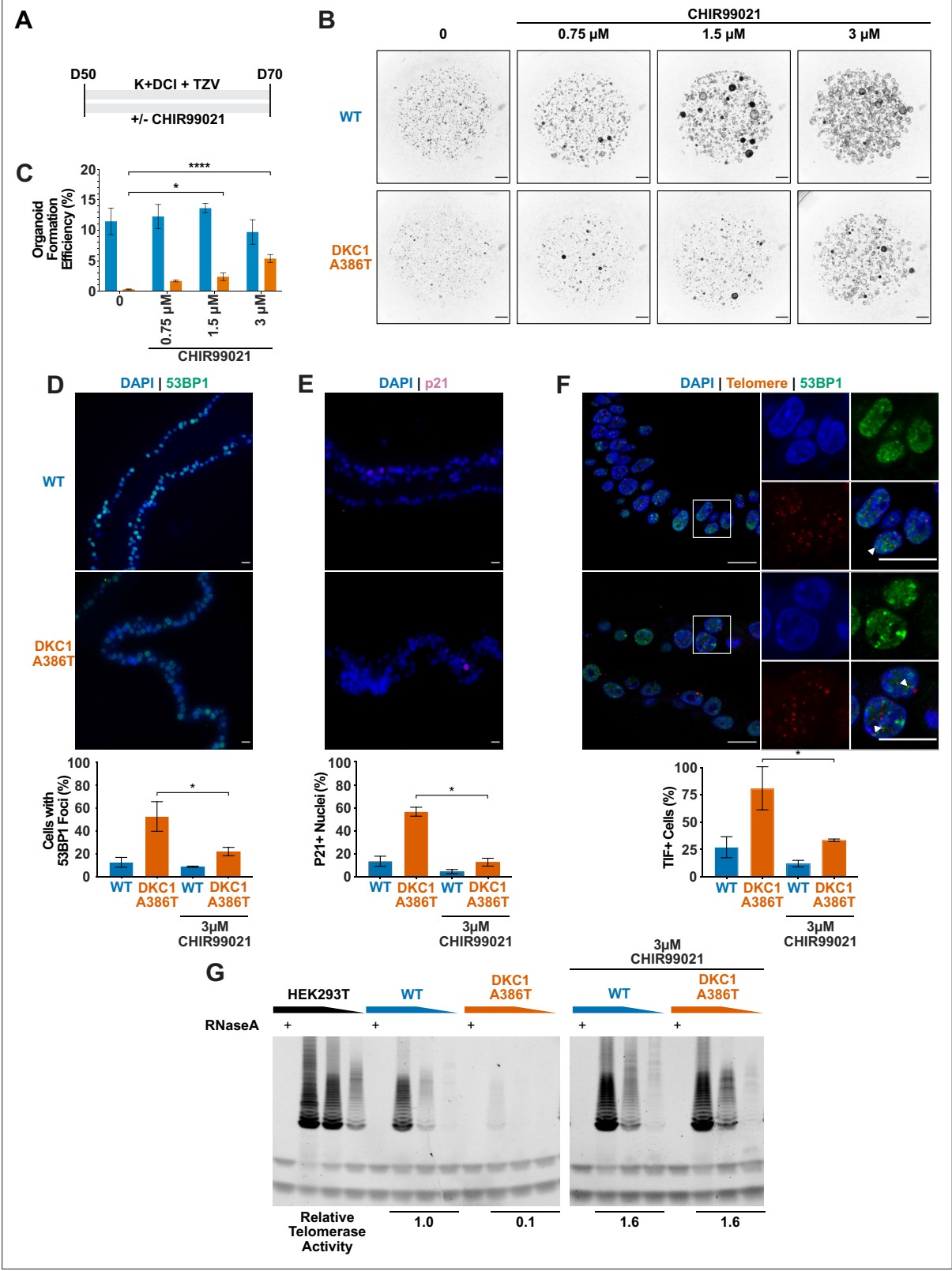

**Figure 4.** CHIR99021 rescues growth and telomere defects in dyskeratosis congentia (DC) induced pluripotent stem cell-derived type II alveolar epithelial (iAT2) cell alveolospheres. (**A**) Differentiation protocol used to test how CHIR99021 affects growth of DC iAT2s. (**B**) Representative images of differentiating wild type (WT) and *DKC1* A386T mutant bearing cells with increasing amounts of CHIR99021 (scale bars, 1 mm for all alveolosphere images). (**C**) Quantifications of alveolosphere formation efficiency after treatment with differing concentrations of CHIR99021 (n = 4, * p<0.05, ****

*Figure 4 continued on next page*

*Figure 4 continued*

p<0.0001, Student's t-test). (**D**) When D70 alveolospheres are cultured with 3 µM CHIR99021, *DKC1* A386T mutant iAT2 cells have a lower fraction of cells with 53 BP1 foci. Note, data for no CHIR99021 bars are from *Figure 2* (n = 3, * p<0.05, Student's t-test; scale bars, 10 µm). (**E**) When D70 alveolospheres are cultured with 3 µM CHIR99021, *DKC1* A386T mutant iAT2 cells have a lower fraction of p21 positive cells. Note, data for no CHIR99021 bars are from *Figure 2* (n = 3, * p<0.05, Student's t-test; scale bars, 10 µm). (**F**) When D70 alveolospheres are cultured with 3 µM CHIR99021, *DKC1* A386T mutant iAT2 cells have a lower fraction of telomere dysfunction induced foci (TIF) positive cells. Note, data for no CHIR99021 bars are from *Figure 2* (n = 3, * p<0.05, Student's t-test; scale bars, 10 µm; insets highlight cells with TIFs, each one noted by the white arrowheads). (**G**) Telomeric repeat amplification protocol assay for telomerase activity in iAT2 cells using fivefold extract dilutions (n = 3).

The online version of this article includes the following source data and figure supplement(s) for figure 4:

**Source data 1.** Source data for *Figure 4C-G*.

**Figure supplement 1.** The growth defect in dyskeratosis congenita (DC) induced pluripotent stem cell-derived type II alveolar epithelial (iAT2) cells is rescued by continuous treatment of CHIR99021 from D28 to D70 but not if withdrawn at D56.

**Figure supplement 2.** CHIR98014 can rescue dyskeratosis congenita (DC) induced pluripotent stem cell-derived type II alveolar epithelial (iAT2) cell growth.

**Figure supplement 3.** TeSLA revealed a modest increase in average telomere length but no apparent decrease in the frequency of shortest detectable telomeres.

**Figure supplement 3—source data 1.** Source data for *Figure 4—figure supplement 3*.

in the mutant iAT2s, the high selectivity of CHIR99021 for GSK3 and its common use as a replacement for Wnt ligands in cell culture differentiation protocols, along with the fact that two different GSK3 inhibitors rescued growth of the mutant iAT2s suggest that the rescue provided by CHIR99021 is indeed related to GSK3 and Wnt (*An et al., 2012*; *Loh et al., 2016*; *Pagliuca et al., 2014*; *Rezania et al., 2014*; *Ring et al., 2003*; *Teo et al., 2014*).

*DKC1* mutant iAT2 cells displayed a senescent phenotype that included upregulation of the p21 cyclin-dependent kinase inhibitor. Recent observations have revealed similar p21 upregulation in the AT2 cells of human PF lungs and have also demonstrated a causal role for p21 in driving experimental PF in mice (*Lee et al., 2021*; *Lv et al., 2022*). Senescence of the mutant iAT2 cells manifested only after serial passaging and telomere shortening, demonstrating that the senescence is not a direct result of *DKC1* mutation per se. The senescence-associated growth defect emerged gradually, becoming more pronounced at later passages. Similarly gradual onset of telomere-driven senescence has been observed in other settings and has been attributed to (i) the stochastic nature of telomere shortening (causing some cells in a population to senesce prior to others) and (ii) graded increases in DNA damage responses in proportion to the extent of telomere shortening (*Cesare et al., 2009*; *Cesare et al., 2013*; *Kaul et al., 2011*; *Martin-Ruiz et al., 2004*; *Ruis and Boulton, 2021*; *Suram and Herbig, 2014*; *Van Ly et al., 2018*).

This Wnt-telomere feedback loop might not be the only regulatory loop at play in AT2 cells. Indeed, we hypothesize that telomeres might be an integrator and amplifier of multiple cellular stress responses. For example, our evidence of upregulation of mitochondrial processes in DC iAT2 cells is consistent with established feedback loops between telomeres and mitochondria: telomere dysfunction can drive mitochondrial dysfunction (*Sahin et al., 2011*) and mitochondrial dysfunction can drive telomere dysfunction (*Guha et al., 2018*; *Passos et al., 2007*; *Qian et al., 2019*). Furthermore, our finding of the upregulation of genes associated with the UPR is consistent with reports linking cellular senescence and the UPR (*Pluquet et al., 2015*). We hypothesize that the UPR might in turn drive telomere dysfunction, and it would thus be of interest to investigate telomere status in familial PF driven by misfolded proteins (*Katzen and Beers, 2020*). These multiple integrated loops might help explain how these various vital cellular processes combine to cause dysfunction in AT2 cells in PF.

Recent work, using mouse AT2 cell organoids, has elucidated a developmental trajectory by which AT2 cells can differentiate via an intermediate state into type I alveolar epithelial (AT1) cells in response to bleomycin injury (*Choi et al., 2020*; *Kobayashi et al., 2020*; *Strunz et al., 2020*). Of note, the intermediate cells are characterized by high levels of p53 signaling and DNA damage which resolve with their final transition to an AT1 cell fate (*Kobayashi et al., 2020*), arguing that repairing DNA damage, potentially at telomeres, is an important step in the transition to an AT1 cell. Furthermore, many of the pathways that promote the differentiation of AT2 cells into AT1 cells are downregulated in DC iAT2 cells, including those involving IL1β, glycolysis, and HIF1α (*Choi et al., 2020*). We speculate that DC AT2 cells with short uncapped telomeres may have trouble suppressing DNA damage at telomeres

as they differentiate into AT1 cells. This may lock these cells into the transition state and may help explain their persistence in PF lungs. These persistent transitioning AT2 cells may thereby contribute to fibrosis. Testing this idea in the human iPSC-derived alveolosphere model will require technical advances to enable the generation of AT1 cells. Regardless, our DC iAT2 cell model recapitulates many hallmarks of PF AT2 cells and offers a new system to probe the underlying biology of PF.

# Materials and methods

See Appendix 1 for Key Resources Table.

## iPSC line generation and maintenance

The BU3 NGST line was a generous gift from Dr. Darrell Kotton at Boston University. iPS cells used for differentiation were maintained on growth factor reduced Matrigel (Corning) coated plates in StemMACS iPS-Brew XF medium (Miltenyi Biotec). Cells were cultured in clusters and passaged every 4–5 days using StemMACS Dissociation reagent (Miltenyi Biotec). All iPS lines were genotyped using an RFLP at the relevant important loci, and the sequence was confirmed by Sanger sequencing. All cells were routinely screened for mycoplasma contamination using a PCR-based assay (*Uphoff and Drexler, 2014*).

## CRISPR editing for generating paired *DKC1* mutant cell lines in BU3 NGST

To generate the introduced BU3 NGST line harboring the *DKC1* mutation, we used the CRISPR track on the UCSC genome browser to select candidate guideRNAs (gRNAs) that targeted as close to the individual mutation as possible, had easily mutable PAM sites and would introduce a new restriction site to make screening easier (see the Key Resources Table for exact sequences and Figure S1 for details). gRNAs were ordered as oligos from IDT and cloned into pX458, a gift from Dr. Feng Zhang's lab (Addgene # 48138). The candidate guides were tested for cutting efficiency by transfecting them into HEK293T cells and assaying cutting efficiency using T7E1 digestion of the PCR amplified locus. The most efficient guides were chosen and ssODN HDR templates were designed to eliminate the PAM. iPS cells were then nucleofected with the Amaxa Nucleofection system using the following program (P3, CA-137) (Lonza). The cells were allowed to recover for 36–48 hr at high density in the presence of ROCK inhibitor and then isolated by FACS for GFPhi cells. They were plated at low density (2500 cells/10 cm plate) and allowed to form single colonies. After 7–10 days, individual clones were selected and transferred to 96 well plates and screened for introduction of the restriction site for each mutation. Restriction enzyme positive clones were expanded and then subjected to sanger sequencing for identification of correctly edited clones. Successfully edited clones were checked for normal karyotype by G-banding (Cell Line Genetics), mycoplasma contamination, and pluripotency marker expression by immunofluorescence and were subsequently passaged for at least five passages before being re-genotyped to ensure that the clones were not mixed. During differentiations, all iAT2 cells were genotyped by restriction digest to ensure that the mutation was not lost with passage.

## Directed differentiation into *NKX2.1+* lung progenitors and *SFTPC+* iPS-derived AT2 cells

A modified version of the protocol described in *Jacob et al., 2017* was used to generate *SFTPC* expressing iAT2 cells. In brief, iPS cells were seeded at 500,000 cells per well on a six-well plate with ROCK inhibitor for 24 hr and incubated at 5% O2 | 5% CO2 | 90% N2. Definitive endoderm was induced using the StemDiff Definitive Endoderm kit for 3 days. Next, the cells were split at a ratio of 1:3 onto fresh Matrigel plates and anteriorized using dorsomorphin (2 µM) and SB431542 (10 µM) in complete serum-free differentiation media (cSFDM) for 3 days. Cells were then differentiated into *NKX2.1+* progenitors by incubating in CBRa media (cSFDM containing CHIR99021 [3 µM], BMP4 [10 ng/mL], and retinoic acid [100 nM]) for 7 days changing media every 2 days at first and then increasing to every day media changes when the media became more acidic. On day 15 or 16, *NKX2.1+* progenitors were isolated using a FACSJazz sorter using the endogenous *NKX2.1::GFP* reporter.

 *NKX2.1+* sorted cells were re-plated at a density of 400,000 cells/mL in 90% Matrigel supplemented with 10% of CK + DCI + TZV media (cSFDM containing 3 µM CHIR99021, 10 ng/mL KGF,

100 nM dexamethasone, 100 µM 8Br-cAMP and 100 µM IBMX, and 2 µM TZV) (from now on referred to as 90/10 Matrigel). The Matrigel droplets were allowed to cure at 37°C for 20–30 min and then overlaid with an appropriate amount of CK + DCI + TZV media. These alveolosphere containing Matrigel droplets were incubated at 37°C at 20% O2 | 5% CO2 | 75% N2 (room air) for 14 days changing with fresh media every other day. Per Hurely et al., CHIR99021 was withdrawn from the media on day 17 for 4 days and re-added back on day 21 to optimize generation and specification of *SFTPC+* cells. On day 28, the iAT2 containing alveolospheres were sorted on a FACSJazz sorter for SFTPC+ cells using the endogenous *SFTPC::TdTomato* reporter. These sorted SFTPC+ cells were replated at a concentration of 65,000 cells/mL in 90/10 Matrigel drops and grown in K + DCI + TZV at 37°C in an ambient air incubator supplemented to 5% CO2 for 3 weeks changing media every other day.

## Alveolosphere counting and formation efficiency calculations

Alveolosphere images were taken on a Leica Thunder widefield microscope using a 1.25× objective. Z-stacks were maximum projected and then thresholded using ImageJ to create a binary file. Binary files were eroded and dilated to ensure maximum determination of the alveolosphere size. Finally, the binary images were separated by watershedding, and alveolospheres were counted using Analyze Particles in ImageJ.

## Immunofluorescence microscopy of iAT2 alveolospheres

Alveolospheres were washed with PBS and then fixed in place using 2% paraformaldehyde (PFA) at room temperature for 30 min and then dehydrated and paraffin embedded and sectioned. Once cut, slides were de-paraffinized, rehydrated, permeabilized, and antigens were retrieved by steaming for 15 min in a citrate buffer (Vector Labs). After blocking, each slide was incubated with a primary antibody using the concentrations listed in the Key Resource Table. Slides were incubated with primary antibody at 37°C for 2 hr. After washing, slides were incubated with appropriate fluorochrome conjugated secondary antibodies (see Key Resources Table for antibody details). Slides were then washed, counterstained with DAPI, and mounted. Images were acquired using a Leica Thunder widefield microscope.

TIFs were stained as described in *Suram et al., 2012*. In brief, cut slides were de-paraffinized, rehydrated, permeabilized, and antigens were retrieved as for other immunofluoresence stains. Slides were blocked and stained for 53BP1 and then stained with an appropriate fluorochrome conjugated secondary antibody. Slides were then re-fixed with PFA, quenched with glycine, re-dehydrated in an ethanol series, and air dried. The slides were then stained with the PNA probe. The slides were washed, rehydrated in an ethanol series, and stained with a tertiary fluorochrome conjugated antibody. Slides were then washed, counterstained with DAPI, and mounted. TIF images were acquired using a Leica SP8 confocal microscope. Quantification of nuclei was carried out in ImageJ in a blinded fashion.

## Measurement of telomerase activity with TRAP

iPS cells or iAT2 cells were cultured as indicated in each figure legend. 100,000 cells were harvested using methods described and lysed using NP-40 lysis buffer and processed as described in *Herbert et al., 2006*. In brief, lysates were incubated with a telomerase substrate and incubated at 30°C for telomerase to add telomere repeats. The reactions were then PCR amplified. Telomere repeats were resolved on a 4–20% TBE polyacrylamide gel and visualized by staining with SYBR Green nucleic acid gel stain. Relative telomerase activity was quantified using ImageJ focusing on the first six amplicons averaged across the dilutions.

## Measurement of telomere lengths by TRF and TeSLA

Telomere lengths were measured as described in *Lai et al., 2016*; *Lai et al., 2017*. DNA was isolated from cells using a Gentra Puregene kit (Qiagen). DNA was quantified by fluorometry using Qubit 2.0 (Invitrogen). For terminal restriction fragment (TRF) analysis in brief, 500 ng of DNA was digested with CviAII overnight followed by digestion with a mixture of BfaI, MseI, and NdeI overnight. For TeSLA in brief, 50 ng of DNA was ligated to telorette adapters, then digested with CviAII, then digested with a combination of BfaI, MseI, and NdeI, dephosphorylated, and TeSLA adapters (AT/TA Adapters) were

ligated on. These TeSLA libraries were PCR amplified using Lucigen's FailSafe polymerase kit with Pre-Mix H.

Southern blotting was carried out using previously established protocols with some modification (*Kimura et al., 2010*; *Lai et al., 2017*). TRFs and TeSLA PCR reactions were separated on a 0.7% agarose gel at 0.833 V/cm for 24 hr. The gel was depurinated and denatured and then transferred to a Hybond XL membrane (Cytiva) by capillary transfer using denaturation buffer. The Hybond membrane was hybridized using a DIG-labeled telomere probe overnight. The blot was then washed and exposed using CDP-Star on an LAS-4000 Image Quant imager (Cytiva). TRFs were analyzed using ImageQuant while TeSLAs were analyzed using the MatLab software developed previously (*Lai et al., 2017*).

## Measurement of telomere lengths by qPCR

Average telomere length was measured by qPCR as described in *Cawthon, 2002*; *Joglekar et al., 2020* with some modifications. In brief, isolated genomic DNA was quantified by Qubit fluorometry (Invitrogen) and diluted to within the range of a standard curve constructed from a mixture of all samples analyzed. Triplicate qPCR reactions of the telomeric (T) product and the single copy gene (S) (*HBB*) were amplified using a Roche LightCycler 480 II (Roche) using the following programs: T PCR program 95°C for 10 min, 40 cycles of 95°C for 15 s, 56°C for 1 min; S PCR program 95°C for 10 min, 40 cycles of 95°C for 15 s, 58°C for 1 min. Cq values were computed using the second derivative method, and T/S ratios were calculated using the $2^{-\Delta\Delta Ct}$ method.

## RNA-sequencing and data analysis

SFTPC + sorted cells from the indicated times during, counted and harvested in TRIzol and stored at –80°C until further processing. The RNA was isolated using a Direct-Zol kit (Zymo Research). RNA concentration was obtained by Qubit fluorometry (Invitrogen), and the integrity was checked by tape station analysis (Agilent Technologies). All samples had RINs >8, and the libraries were prepared by poly-A selection and sequenced by GeneWiz, LLC.

RNA sequencing libraries were prepared using the NEBNext Ultra RNA Library Prep Kit for Illumina following manufacturer's instructions (NEB). Briefly, mRNAs were first enriched with oligo (dT) beads. Enriched mRNAs were fragmented for 15 min at 94°C. First strand and second strand cDNAs were subsequently synthesized. cDNA fragments were end repaired and adenylated at 3' ends, and universal adapters were ligated to cDNA fragments, followed by index addition and library enrichment by limited-cycle PCR. The sequencing libraries were validated on the Agilent TapeStation (Agilent Technologies), and quantified using a Qubit 2.0 Fluorometer (Invitrogen) as well as by quantitative PCR (KAPA Biosystems).

The sequencing libraries were pooled and clustered on one lane of a flowcell. After clustering, the flowcell was loaded on the Illumina HiSeq instrument (4000 or equivalent) according to manufacturer's instructions. The samples were sequenced using a 2 × 150 bp paired end configuration. Image analysis and base calling were conducted by the HiSeq control software. Raw sequence data (.bcl files) generated from Illumina HiSeq was converted into fastq files and de-multiplexed using Illumina's bcl2fastq 2.17 software. One mismatch was allowed for index sequence identification.

Fastq files were checked for quality using FastQC. Raw sequence files (fastq) for 22 samples were mapped using salmon (https://combine-lab.github.io/salmon/; *Patro et al., 2017*) against the human transcripts described in Gencode (version v33, built on the human genome GRCm38, https://www.gencodegenes.org), with a 70.5% average mapping rate yielding 30.4 M average total input reads per sample. Transcript counts were summarized to the gene level using tximport (available here) and normalized and tested for differential expression using DESeq2 (available here). Subsets of time-matched samples were used to compute pair-wise contrast statistics for mutant vs WT at each time. Raw FASTQ files and analyzed counts are available on GEO under the accession number GSE160871.

GSEA (*Subramanian et al., 2005*) was carried out in R (v4.0.2) (*R Development Core Team, 2020*) using RStudio (v1.3.1056) (*RStudio Team, 2020*), the tidyverse (v1.3.0) (*Wickham et al., 2019*), and the readxl package (v1.3.1) (*Hadley Wickham, 2019*). GSEA was run for contrasts of interest in preranked mode using the DESeq2 statistic as the ranking metric (*Love et al., 2014*). Annotated molecular signatures from the Hallmark Collection (H), Curated Collection (C2), and Regulatory Target Molecular Collection (C3) maintained by the Molecular Signatures Database were accessed in RStudio using the msigdbr package (v7.1.1) (*Dolgalev, 2020*; *Liberzon et al., 2011*; *Liberzon et al., 2015*).

The clusterProfiler package (v3.17.1) was used to perform GSEA on the unfiltered, sorted gene lists (*Yu et al., 2012*). GSEA results were viewed using the DT package (v0.15) (*Yihui Xie, 2020*). GSEA plots were generated using the enrichplot package (v1.9.1) (*Yu, 2020*).

Genes that differed in expression >twofold and were associated with an adjusted p-value <0.05 from the D70 time point were also analyzed through the use of IPA (Ingenuity Systems, https://www. ingenuity.com) (*Krämer et al., 2014*) through the University of Pennsylvania Molecular Profiling Facility.

## ddPCR

RNA was isolated as listed above for the RNA-sequencing. 140 ng of RNA was first treated with ezDNAse (Invitrogen) and converted to cDNA using the SuperScript IV First-Strand Synthesis System (Invitrogen). The equivalent of 12.5 ng of original RNA was used as template in the reactions, and the ddPCR assays as well as the ddPCR supermix used are listed in the Key Resources Table (Bio-Rad). ddPCR was performed according to manufacturer's instructions using QX200 AutoDG Droplet Digital PCR System (Bio-Rad). Results were analyzed using Quanta-Soft Version 1.7.4.0917 (Bio-Rad). Copies/ μL were used and normalized to *ACTB*.

## Statistical methods

Statistical methods are outlined in each of the figure legends. Each replicate 'n' represents an entirely separate differentiation from the pluripotent stem cell stage. For experiments with iPS cells, 'n' represents separate wells of the same iPS line (specifically for Figure S1 and S2). Quantitative data is represented as the mean with error bars representing the standard error of the mean. Student's t-tests (unpaired and two-tailed) were used for determining statistical significance for all comparisons unless otherwise noted.

## Acknowledgements

We would like to thank Drs. Eric Witze, Peter Klein, Timothy Olson, Andrew Vaughan, Leo Cardenas, and Aravind Sivakumar for fruitful discussions. We would like to thank Dr. Dong Hun Woo, Carla Hoge, John McCormick, and Dr. Rachel Truitt for help culturing iPS cells, and Dr. Darrell Kotton for providing the BU3 NGST line. We also thank Drs. Erica Carpenter and Jacob Till for help with the ddPCR. This work was also supported in part by the National Institute of Diabetes and Digestive and Kidney Diseases (NIDDK) Center for Molecular Studies in Digestive and Liver Diseases (P30DK050306) and its Molecular Profiling and Imaging core facilities, by the University of Pennsylvania induced Pluripotent Stem Cell Core and the Penn Genomic Analysis Core, by the Flow Cytometry Core Laboratory, the Human Pluripotent Stem Cell Core at the Children's Hospital of Philadelphia Research Institute, and the Abramson Cancer Center BioRepository Shared Resource and the Cancer Center Support Grant from the NCI, P30 CA016520.

## Additional information

### Competing interests

Edward E Morrisey: Reviewing editor, *eLife*. The other authors declare that no competing interests exist.

### Funding

| Funder | Grant reference number | Author |
| --- | --- | --- |
| National Institute on Aging | R21AG054209 | Christopher J Lengner<br>F Brad Johnson |
| National Institute on Aging | 5T32AG000255 | Rafael Jesus Fernandez |
| Team Telomere/Penn Orphan Disease Center | | Christopher J Lengner<br>F Brad Johnson |

| Funder | Grant reference number | Author |
|---|---|---|
| National Heart, Lung, and Blood Institute | R01HL148821 | Christopher J Lengner F Brad Johnson |

The funders had no role in study design, data collection and interpretation, or the decision to submit the work for publication.

#### Author contributions
Rafael Jesus Fernandez, Conceptualization, Data curation, Formal analysis, Investigation, Methodology, Project administration, Visualization, Writing – original draft, Writing – review and editing; Zachary JG Gardner, Data curation, Formal analysis, Investigation, Methodology, Visualization; Katherine J Slovik, Methodology; Derek C Liberti, Methodology, Writing – review and editing; Katrina N Estep, Qijun Chen, Garrett T Santini, Javier V Perez, Sarah Root, Ranvir Bhatia, Investigation; Wenli Yang, Methodology, Resources; John W Tobias, Data curation, Formal analysis, Investigation, Methodology, Resources, Software, Visualization; Apoorva Babu, Michael P Morley, Data curation, Formal analysis, Resources, Software, Visualization; David B Frank, Edward E Morrisey, Resources; Christopher J Lengner, Conceptualization, Investigation, Methodology, Project administration, Resources, Supervision, Writing – review and editing; F Brad Johnson, Conceptualization, Funding acquisition, Investigation, Methodology, Project administration, Resources, Supervision, Writing – original draft, Writing – review and editing

#### Author ORCIDs
Rafael Jesus Fernandez  http://orcid.org/0000-0001-9295-4810
Derek C Liberti  http://orcid.org/0000-0003-2991-9283
Edward E Morrisey  http://orcid.org/0000-0001-5785-1939
Christopher J Lengner  http://orcid.org/0000-0002-0574-5189
F Brad Johnson  http://orcid.org/0000-0002-7443-7227

#### Decision letter and Author response
Decision letter https://doi.org/10.7554/eLife.64430.sa1
Author response https://doi.org/10.7554/eLife.64430.sa2

## Additional files

#### Supplementary files
• Supplementary file 1. Gene set enrichment analysis (GSEA) results comparing D70 DC to wild type (WT) iPSC-derived type II alveolar epithelial (iAT2) cells. These tables provide the unedited output of the GSEA analysis using the C2 curated gene sets, H hallmark gene sets, and C3 regulatory target gene sets when comparing D70 mutant to WT iAT2 cells. The table reports the name of the gene set (ID), the size of the gene set (setSize), the raw enrichment score (enrichmentScore), the normalized enrichment score, along with the p-value, the adjusted p-value (p.adjust) and false discovery rate q-value (q-values). The 'core enrichment' column displays the genes in the gene set.

• Supplementary file 2. Ingenuity pathway analysis (IPA) results comparing D70 dyskeratosis congenita to wild type iPSC-derived type II alveolar epithelial (iAT2) cells. These tables provide the unedited output of the IPA. The summary tab lists metadata associated with the analysis. The 'analysis ready molecules' lists the differentially expressed genes that differed in expression greater than twofold and were associated with an adjusted p-value < 0.05 from D70 mutant iAT2 cells (DEG list). The 'canonical pathways tab' lists pathways curated from the literature and the p-value for enrichment using the DEG list used in the IPA. The 'upstream regulators' tab lists the transcription factors, cytokines, and other genetic regulators whose target genes are in the DEG list. 'Causal networks' seeks to build a regulatory network based off of the 'upstream regulators' to identify master regulators of the DEG list. For more information on interpreting ingenuity analysis results see *Krämer et al., 2014*.

• Supplementary file 3. Differentially expressed pathways from D70 dyskeratosis congenita iPSC-derived type II alveolar epithelial (iAT2) cells that are similar to changes seen in pulmonary fibrosis. These tables display selected results from gene set enrichment analysis (GSEA) and ingenuity pathway analyses (IPAs) that highlight pathways found to be differentially regulated in mutant iAT2 cells at D70 when compared with wild type cells. The first table displays GSEA results along with the

pathway name, normalized enrichment score, and adjusted p-value (D70 p-adj). The second table displays IPA results from the canonical pathways analysis. These are gene sets that are differentially regulated in mutant iAT2 cells at D70 when compared with wild type cells. The p-value reports the significance of enrichment of the molecules in that gene set, and the activation score reports how concordant the gene expression changes are with what is predicted from the literature embedded in IPA (a negative z-score argues that the gene set is down regulated in the mutant iAT2 cells, whereas a positive z-score argues that the gene set is upregulated in mutant iAT2 cells; the lack of a z-score is indicative there was insufficient evidence to provide a z-score.) The 'molecules' column lists the genes that were in that gene set that were also found in our differentially expressed gene list when comparing mutant iAT2 cells to wildtype cells.

• Transparent reporting form

### Data availability
Sequencing data was deposited in GEO: GSE160871.

The following dataset was generated:

| Author(s) | Year | Dataset title | Dataset URL | Database and Identifier |
|---|---|---|---|---|
| Fernandez RJ, Johnson FB, Tobias JW | 2021 | Transcriptional profiling of short telomere iPS-derived sorted SFTPC+ iAT2 cells | https://www.ncbi.nlm.nih.gov/geo/query/acc.cgi?acc=GSE160871 | NCBI Gene Expression Omnibus, GSE160871 |

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

# Appendix 1

## Key resources table

### Appendix 1—key resources table

| Reagent type (species) or resource | Designation | Source or reference | Identifiers | Additional information |
|---|---|---|---|---|
| Cell line (*Homo sapiens*) | BU3 NGST | *Jacob et al., 2017* | | iPS cell line |
| Cell line (*Homo sapiens*) | BU3 NGST – WT | This Paper | | iPS cell line, WT control passaged with the DKC1 A386T cell line. See *Figure 1—figure supplement 1* & 2 for more details. Contact the corresponding authors for the cell line. |
| Cell line (*Homo sapiens*) | BU3 NGST - DKC1 A386T | This Paper | | iPS cell line, DKC1 A386T passaged with the WT cell line. See *Figure 1—figure supplement 1* & 2 for more details. Contact the corresponding authors for the cell line. |
| Cell line (*Homo sapiens*) | AG04646 | *Woo et al., 2016* | | iPS cell line |
| Recombinant DNA reagent | pX458 | Feng Zhang | Addgene # 48,138 | |
| Chemical compound, drug | Thiazovivin (TZV) | Cayman Chemical compound, drug | CAT# 14,245 | |
| Chemical compound, drug | SB431542 | Cayman Chemical compound, drug | CAT# 13,031 | |
| Chemical compound, drug | Dorsomorphin | Cayman Chemical compound, drug | CAT# 21,207 | |
| Chemical compound, drug | CHIR99021 | Cayman Chemical compound, drug | CAT# 13,122 | |
| Chemical compound, drug | CHIR98014 | Cayman Chemical compound, drug | CAT# 15,578 | |
| Chemical compound, drug | Retinoic Acid | Sigma | CAT# R2625-50MG | |
| Peptide, recombinant protein | rhBMP4 | R&D Systems | CAT# 314 BP-050 | |
| Peptide, recombinant protein | rhKGF | R&D Systems | CAT# 251 KG-050 | |
| Chemical compound, drug | Dexamethasone | Sigma | CAT# D4902-100MG | |
| Chemical compound, drug | 8Br-cAMP | Sigma | CAT# 7880–250 MG | |
| Chemical compound, drug | IBMX | Sigma | CAT# I5879-250MG | |
| Peptide, recombinant protein | DNaseI | Roche | CAT# 4716728001 | |
| Chemical compound, drug | 7-AAD | BD Biosciences | CAT# 51-68981E | |
| Chemical compound, drug | Propidium Iodide | BD Biosciences | CAT# 556,463 | |
| Peptide, recombinant protein | Dispase II | Gibco | CAT# 17105041 | |
| Chemical compound, drug | SYBR Green I Nucleic Acid Gel Stain | Invitrogen | CAT# S7563 | |
| Peptide, recombinant protein | Matrigel, Growth Factor Reduced | Corning | CAT# 354,230 | |

*Appendix 1 Continued on next page*

*Appendix 1 Continued*

| Reagent type (species) or resource | Designation | Source or reference | Identifiers | Additional information |
|---|---|---|---|---|
| Chemical compound, drug | Stem MACS iPS-Brew XF | Miltenyi Biotec | CAT# 130-104-368 | |
| Peptide, recombinant protein | Accutase | Innovative Cell Technologies | CAT# AT104-500 | |
| Commercial assay, kit | P3 Primary Cell 4D-NucleofectorTM X Kit L | Lonza | CAT# V4XP-3012 | |
| Chemical compound, drug | IMDM | Corning/Mediatech | CAT# MT10-016-CV | |
| Chemical compound, drug | Ham's F12 Nutrient Mix | Corning/Mediatech | CAT# MT10-080-CV | |
| Chemical compound, drug | B27 | Gibco | CAT# 17504044 | |
| Chemical compound, drug | N2 | Gibco | CAT# 17502048 | |
| Chemical compound, drug | BSA Fraction V | Gibco | CAT# 15260037 | |
| Chemical compound, drug | Monothioglycerol | Sigma | CAT# M6145-25ML | |
| Chemical compound, drug | GlutaMAX | Gibco | CAT# 35050061 | |
| Chemical compound, drug | Ascorbic Acid | Sigma | CAT# A4544-25G | |
| Chemical compound, drug | Primocin | Invivogen | CAT# ANT-PM-2 | |
| Chemical compound, drug | Hank's Balanced Salt Solution | Gibco | CAT# 14175079 | |
| Chemical compound, drug | HEPES | Gibco | CAT# 15630080 | |
| Chemical compound, drug | EDTA | Gibco | CAT# 15575039 | |
| Peptide, recombinant protein | 0.05% trypsin-EDTA | Gibco | CAT# 25300054 | |
| Commercial assay, kit | Direct-zol RNA Micro Kit | Zymo | CAT# R2061 | |
| Commercial assay, kit | Gentra Puregene Cell Kit | Qiagen | CAT# 158,745 | |
| Antibody | Anti-NANOG (Rabbit polyclonal) | Reprocell/Stemgent | CAT# 09–0020 | IF (1:100 37°C for 2 hr) |
| Antibody | Anti-53BP1 (Rabbit monoclonal) | Novus | CAT# NB100-304 | IF (1:100 37°C for 2 hr) |
| Antibody | Anti-P21 (Mouse monoclonal) | Santa Cruz | CAT# sc-6246 | IF (1:50 37°C for 2 hr) |
| Antibody | Anti-Cleaved Caspase 3 (CC3) (Rabbit monoclonal) | Cell Signaling Technologies | CAT# 9,664 | IF (1:2000 37°C for 2 hr) |
| Sequence-based reagent | Anti-Cy3-Telo-C Probe | PNA Bio | CAT# F1002 | Anti-telomeric |
| Antibody | Anti-Digoxigenin-AP, Fab fragments | Roche | CAT# 11093274910 | |
| Antibody | Goat anti-Rabbit IgG Secondary Antibody, Alexa Fluor 647 (Donkey polyclonal) | Invitrogen | CAT# A-21244 | IF (1:250 RT for 1 hr) |
| Antibody | Donkey anti-Mouse IgG Secondary Antibody, Alexa Fluor 647 (Donkey polyclonal) | Invitrogen | CAT# A-31571 | IF (1:250 RT for 1 hr) |
| Antibody | Donkey anti-Goat IgG Secondary Antibody, Alexa Fluor 647 (Donkey polyclonal) | Invitrogen | CAT# A-21447 | IF(1:250 RT for 1 hr) |

*Appendix 1 Continued on next page*

*Appendix 1 Continued*

| Reagent type (species) or resource | Designation | Source or reference | Identifiers | Additional information |
|---|---|---|---|---|
| Chemical compound, drug | TriZOL | Invitrogen | CAT# 15596018 | |
| Peptide, recombinant protein | FailSafe Polymerase Enzyme Mix | Lucigen | CAT# FS9901K | |
| Chemical compound, drug | FailSafe Buffer H | Lucigen | CAT# FSP995H-INCL | |
| Peptide, recombinant protein | CviAII | NEB | CAT# R0640L | |
| Peptide, recombinant protein | BfaI | NEB | CAT# R0568L | |
| Peptide, recombinant protein | MseI | NEB | CAT# R0525L | |
| Peptide, recombinant protein | NdeI | NEB | CAT# R0111L | |
| Peptide, recombinant protein | rSAP | NEB | CAT# M0371L | |
| Chemical compound, drug | CDP-Star | Roche | CAT# 11759051001 | |
| Chemical compound, drug | DIG-11-dUTP | Roche | CAT# 11558706910 | |
| Chemical compound, drug | DIG Easy Hybridization Granules | Roche | CAT# 11796895001 | |
| Chemical compound, drug | Blocking Reagent | Roche | CAT# 11096176001 | |
| Sequence-based reagent | DIG Labeld DNA Molecular Weight Marker II | Roche | CAT# 11218590910 | |
| Chemical compound, drug | Hybond-XL | Cytiva/Amersham | CAT# RPN303S | |
| Peptide, recombinant protein | Klenow Fragment | NEB | CAT# M0212S | |
| Peptide, recombinant protein | Lambda Exonuclease | NEB | CAT# M0262S | |
| Peptide, recombinant protein | Go-TAQ Flexi | Promega | CAT# M8298 | |
| Peptide, recombinant protein | ddPCR Multiplex Supermix | BioRad | CAT# 12005909 | |
| Software | FIJI | NIH | RRID:SCR_002285 | https://imagej.net/Fiji |
| Software | Salmon | *Patro et al., 2017* | | |
| Software | Ingenuity pathway analysis | *Krämer et al., 2014* | | |
| Software | FlowJo | BD Biosciences | RRID:SCR_008520 | |
| Software | ImageQuant TL 8.2 | Cytiva | RRID:SCR_018374 | |
| Software | TeSLAQuant | *Lai et al., 2017* | | |

## Appendix 2

## Primer Sequences

| Reagent type (species) or resource | Designation | Source or reference | Identifiers | Additional information |
|---|---|---|---|---|
| Sequence-based reagent | gRNA-DKC1A386-5-F | This Paper | gRNA Cloning Primers designed to target DKC1 A386 – See *Figure 1—figure supplement 1* for details | CAC CGC TTC TTC TGA CTT GCC TGG |
| Sequence-based reagent | gRNA-DKC1A386-5-R | This Paper | gRNA Cloning Primers designed to target DKC1 A386 – See Figure 1—figure supplement 1 for details | AAA CCC AGG CAA GTC AGA AGA AGC |
| Sequence-based reagent | DKC1-Geno-F | *Woo et al., 2016* | Genotyping primers for DKC1 Gene | AAA AGT AGA GTG GCT GCG GT |
| Sequence-based reagent | DKC1-Geno-R | *Woo et al., 2016* | Genotyping primers for DKC1 Gene | ACA ACC TCC ATG CTC ACC TG |
| Sequence-based reagent | TeSLA-T1 | *Lai et al., 2017* | Telorette | ACT GGC CAC GTG TTT TGA TCG ACC CTA AC |
| Sequence-based reagent | TeSLA-T2 | *Lai et al., 2017* | Telorette | ACT GGC CAC GTG TTT TGA TCG ATA ACC CT |
| Sequence-based reagent | TeSLA-T3 | *Lai et al., 2017* | Telorette | ACT GGC CAC GTG TTT TGA TCG ACC TAA CC |
| Sequence-based reagent | TeSLA-T4 | *Lai et al., 2017* | Telorette | ACT GGC CAC GTG TTT TGA TCG ACT AAC CC |
| Sequence-based reagent | TeSLA-T5 | *Lai et al., 2017* | Telorette | ACT GGC CAC GTG TTT TGA TCG AAA CCC TA |
| Sequence-based reagent | TeSLA-T6 | *Lai et al., 2017* | Telorette | ACT GGC CAC GTG TTT TGA TCG AAC CCT AA |
| Sequence-based reagent | TeSLA Adapter Short | *Lai et al., 2017* | | GGT TAC TTT GTA GCT CTG TC[SpcC3] |
| Sequence-based reagent | TeSLA Adapter TA | *Lai et al., 2017* | | [Phos] TAG ACA GGC TTA CAA AGT AAC CAT GGT GGA GAA TTC TGT CGT CTT CAC GCT ACA TT [SpcC3] |

*Continued on next page*

*Continued*

| Reagent type (species) or resource | Designation | Source or reference | Identifiers | Additional information |
|---|---|---|---|---|
| Sequence-based reagent | TeSLA Adapter AT | *Lai et al., 2017* | | [Phos] ATG ACA GGC TTA CAA AGT AAC CAT GGT GGA GAA TTC TGT CGT CTT CAC GCT ACA TT [SpcC3] |
| Sequence-based reagent | AP | *Lai et al., 2017* | | TGT AGC GTG AAG ACG ACA GAA |
| Sequence-based reagent | TeSLA-TP | *Lai et al., 2017* | | TGG CCA CGT GTT TTG ATC GA |
| Sequence-based reagent | G-Rich ONT | *Lai et al., 2016* | | [Phos] CCC TAA CCC TAA CCC TAA CCC TAA CCC TAA CCC TAG ATA GTT GAG AGT C |
| Sequence-based reagent | Branched Universal Sequence-based reagent | *Lai et al., 2016* | | [Phos] GAC TCT CAA CTA TC +T + A |
| Sequence-based reagent | TS | *Herbert et al., 2006* | | AAT CCG TCG AGC AGA GTT |
| Sequence-based reagent | ACX | *Herbert et al., 2006* | | GCG CGG CTT ACC CTT ACC CTT ACC CTA ACC |
| Sequence-based reagent | Tel1B | Luo et al. 2020 | | CGG TTT GTT TGG GTT TGG GTT TGG GTT TGG GTT TGG GTT |
| Sequence-based reagent | Tel2B | Luo et al. 2020 | | GGC TTG CCT TAC CCT TAC CCT TAC CCT TAC CCT TAC CCT |
| Sequence-based reagent | HBG1 | *Cawthon, 2002* | | GCT TCT GAC ACA ACT GTG TTC ACT AGC |
| Sequence-based reagent | HBG2 | *Cawthon, 2002* | | CAC CAA CTT CAT CCA CGT TCA CC |
| Sequence-based reagent | TINF2 | BioRad | dHsaCPE5027564 | |
| Sequence-based reagent | POT1 | BioRad | dHsaCPE50447370 | |
| Sequence-based reagent | TERT | BioRad | dHsaCPE5048434 | |
| Sequence-based reagent | TERC | BioRad | dHsaCNS923142164 | |
| Sequence-based reagent | ACTB | BioRad | dHsaCPE5190200 | |

