## [Editor Report]

This study reports a new cell line model for Dyskeratosis congenita. The authors characterized this model by examining its ability to form organoids and its defects in telomeres. Transcriptomics analysis unveiled defects in Wnt signaling, and treatment with a Wnt agonist could rescue these defects and enhance telomerase activity. Overall, the study is well designed and executed, the data presented are clear and convincing and the conclusions should be of great interest to researchers studying Dyskeratosis congenita and telomere biology.

---

## [Decision Letter]

**Decision letter after peer review:**

Thank you for submitting your article "GSK3 inhibition rescues growth and telomere dysfunction in dyskeratosis congenita iPSC-derived type II alveolar cells" for consideration by *eLife*. Your article has been reviewed by 2 peer reviewers, one of whom is a member of our Board of Reviewing Editors, and the evaluation has been overseen by Matt Kaeberlein as the Senior Editor. The reviewers have opted to remain anonymous.

Essential revisions:

1) The mutant cells showed down-regulation of shelterin component genes (TINF2 and POT1) downstream of the Wnt signaling pathway. To show that the GSK3 inhibitor CHIR99021 acted through Wnt signaling in iAT2 cells, the authors should show that the expression of these genes can be restored by the treatment.

2) the authors use iPSCs that are 5 vs 25 passages after introduction (or not) of the DKC1 A386T mutation for the generation of iAT2 cells. The data presented in Figure S3D simply does not correspond to the statement that "using iPSCs only five passages after the introduction of the mutation yielded iAT2s without any defects in alveolosphere growth or size at D70 (Figure S3C)" as the images in Figure S3D appear clearly distinguishable. the iPSCs 5 passages after gene editing may certainly be less effected, which would not take away from the authors may point (it is not the DKC1 A386T mutation per se). The manuscript needs to either include rigorous data as in Figure 1C and/or 1D or render a qualitive assessment that is better supported by the data presented.

3) The authors show iAT2 DKC1 mutant organoids generated from the later passage iPSCs have an apparent growth defect as early as Day 50 but that those generated from the earlier passage iPSCs do not at Day 70 [with caveats the images are of different quality (comparing Figure 1B and Figure S3D) and quantitative data (similar to Figure 1C) are lacking for the iAT2 organoids generated from the early passage iPSCs]. This needs to be addressed. Is it possible that there are two different processes – an initial constraint imposed by shortened telomeres (D50) and then a more catastrophic effect due to uncapping (D70)?

4) It is striking that prolonged CHIR99021 treatment (ie, through to Day 70) resulted in increased telomerase activity, and more so in mutant compared to wild type cells. First, how reproducible was this effect? Given TERT expression does not rescue DKC1 mutants but TERC does, were TERC levels increased? Also, given this robust increase, its striking that no difference is detected in TeSLA assays given the proportion of very short detected telomeres that would presumably be substrates for telomerase.

*Reviewer #1 (Recommendations for the authors):*

– The analysis shown in Figure 1C should be done for the experiment performed with iPSCs 5 passages after introduction of the DKC1 A386T mutation.

– An apoptosis positive control is lacking for Figure S3E.

– The figure legend for Figure S4B does not make sense nor correlate with the main text description (lines 209-211). It appears that an experiment was designed whereby CHIR99021 was either present or removed after D56 and the cells analyzed at D70 (Figure S4A) but the data are not shown in Figure S4B.

– The graphics Figure 1A and S3C show Day 49 but the text reads Day 50 – this is confusing.

– It is not stated whether the analyses performed in Figure 2 are using cells derived from iPSCs 25 passages after introduction of the DKC1 mutation (It is likely this is the case, but it should be clearly stated).

– Line 138 – Should refer to Figure S3C-D.

– The methods section should be carefully reviewed for formatting consistency throughout – e.g.., use of capitalization or not.

*Reviewer #2 (Recommendations for the authors):*

1. The mutant cells showed down-regulation of shelterin component genes (TINF2 and POT1) downstream of the Wnt signaling pathway. To show that the GSK3 inhibitor CHIR99021 acted through Wnt signaling in iAT2 cells, the authors should show that the expression of these genes can be restored by the treatment.

2. Is Wnt signaling required for CHIR99021 to show its effects on cellular senescence and telomere dysfunction? This should be examined and discussed.

3. The authors showed little change in telomere length, but increased telomerase activity by CHIR99021 treatment in iAT2 cells. Some further discussion on this will be helpful for readers to understand why this is the case.

4. Wnt has been shown to regulate TERT expression. Hence, it should be tested here whether increased telomerase activity is attributed to increased TERT expression and/or TERC expression.

---

## [Author Response]

Essential revisions:1) The mutant cells showed down-regulation of shelterin component genes (TINF2 and POT1) downstream of the Wnt signaling pathway. To show that the GSK3 inhibitor CHIR99021 acted through Wnt signaling in iAT2 cells, the authors should show that the expression of these genes can be restored by the treatment.

We agree that this is an interesting issue and that demonstrating restoration of shelterin expression by CHIR would support the hypothesis that changes in shelterin levels are involved in the mechanism by which CHIR rescues defects in the mutant iAT2 cells. We have attempted to demonstrate this using two approaches*, but unfortunately have been unable to do so. We are sorry about this (and want to emphasize that it has been very difficult to make these attempts under the conditions of the pandemic, particularly given the months-long cultures needed to do experiments along with unpredictable lab shut downs, as well as supply chain issues (particularly Matrigel shortages)). Regardless, we want to emphasize three things: 1) we haven’t claimed that shelterin upregulation is involved in rescue, 2) whether or not we know the precise mechanism of rescue doesn’t affect the overall findings of our paper nor the potential therapeutic relevance of the rescue, and 3) we now provide new and clearer data supporting a role for differences in telomerase expression and activity (which are well-known to be upregulated by Wnt in diverse contexts) as contributors to mutant defects and to the rescue by CHIR (see Point 4, below).

­­– In particular, we compared mRNA transcript levels in wild type and mutant iAT2 cultures under basal and rescued conditions using both scRNAseq and ddPCR assays. Unfortunately, we had technical problems with the scRNAseq and have so far been unable to repeat the experiment. The ddPCR data was too variable under rescued conditions to be interpretable, and we ran out of RNA before getting satisfactory results (but fortunately the ddPCR did provide clear data comparing wild type and mutant cultures under basal conditions, now provided in Figure 3—figure supplement 1). We will eventually repeat the scRNAseq, but we feel such analyses are really outside the scope of the manuscript.

2) The authors use iPSCs that are 5 vs 25 passages after introduction (or not) of the DKC1 A386T mutation for the generation of iAT2 cells. The data presented in Figure S3D simply does not correspond to the statement that "using iPSCs only five passages after the introduction of the mutation yielded iAT2s without any defects in alveolosphere growth or size at D70 (Figure S3C)" as the images in Figure S3D appear clearly distinguishable. the iPSCs 5 passages after gene editing may certainly be less effected, which would not take away from the authors may point (it is not the DKC1 A386T mutation per se). The manuscript needs to either include rigorous data as in Figure 1C and/or 1D or render a qualitive assessment that is better supported by the data presented.

We appreciate these concerns and have updated the manuscript to offer a qualitative assessment that more accurately reflects the data. Furthermore, we quantified the organoid formation efficiency for the iAT2s derived from passage 5 iPSCs (see Figure 1—figure supplement 4 A-C).

In particular, the modified sentence reads as follows: “In contrast, using iPSCs only five passages after the introduction of the mutation yielded iAT2s with less dramatic defects in alveolosphere growth and size at D70 (Figure S3C-E).”

3) The authors show iAT2 DKC1 mutant organoids generated from the later passage iPSCs have an apparent growth defect as early as Day 50 but that those generated from the earlier passage iPSCs do not at Day 70 [with caveats the images are of different quality (comparing Figure 1B and Figure S3D) and quantitative data (similar to Figure 1C) are lacking for the iAT2 organoids generated from the early passage iPSCs]. This needs to be addressed. Is it possible that there are two different processes – an initial constraint imposed by shortened telomeres (D50) and then a more catastrophic effect due to uncapping (D70)?

We agree that there is a progressive defect with passage. This observation is consistent with a substantial body of literature in the telomere field, and is likely explained by two phenomena:

First, although the rate of telomere shortening is gradual at the level of a population of cells (each with its own collection of 92 telomeres), this rate is somewhat stochastic for any given telomere (e.g. a telomere can shorten dramatically and instantaneously due to an error during DNA replication), leading some cells to begin experiencing growth restriction earlier than other cells. Thus, at the level of a population of cells, growth declines in gradual fashion as an increasing fraction of the population becomes senescent. E.g. see Stochasticity of Telomere Shortening dictates Culture Senescence in Fibroblasts (Martin-Ruiz et al., 2004) and The replicometer is broken: telomeres activate cellular senescence in response to genotoxic stresses (Suram and Herbig, 2014).

Second, there is clear evidence that telomeres do not lose their functions in an “all or none” fashion as they shorten. For example, partially shortened telomeres can activate ATM (and thus senescence) prior to the stage at which telomeres that have shortened even further become fusogenic. It is also thought (but not yet proven) that as telomeres shorten below a particular threshold length, the tendency to activate DNA damage responses may increase in a somewhat graded fashion with additional shortening (e.g. perhaps by impacting the equilibrium between telomeres in linear vs. t-loop conformations). Therefore, it is conceivable that gradual telomere shortening may increasingly inhibit cell cycle progression. E.g. see (Cesare et al., 2009, 2013; Kaul et al., 2011; Ruis and Boulton, 2021a; Van Ly et al., 2018a).

We have modified the Discussion to highlight our thoughts on these points with a new text, as follows:

“Senescence of the mutant iAT2 cells displayed a senescent phenotype that manifested only after serial passaging and telomere shortening, demonstrating that the senescence is not a direct result of *DKC1* mutation *per se*. The senescence-associated growth defect emerged gradually, becoming more pronounced at later passages. Similarly gradual onset of telomere-driven senescence has been observed in other settings, and has been attributed to i) the stochastic nature of telomere shortening (causing some cells in a population to senesce prior to others) and ii) graded increases in DNA damage-responses in proportion to the extent of telomere shortening (Cesare et al., 2009, 2013; Kaul et al., 2011; Martin-Ruiz et al., 2004; Ruis and Boulton, 2021b; Suram and Herbig, 2014; Van Ly et al., 2018b).”

4) It is striking that prolonged CHIR99021 treatment (ie, through to Day 70) resulted in increased telomerase activity, and more so in mutant compared to wild type cells. First, how reproducible was this effect? Given TERT expression does not rescue DKC1 mutants but TERC does, were TERC levels increased? Also, given this robust increase, its striking that no difference is detected in TeSLA assays given the proportion of very short detected telomeres that would presumably be substrates for telomerase.

We agree that this finding was striking (and surprising), and upon repetition we noticed that the WT organoids were at a much higher level of confluence than the *DKC1* mutant organoids when they were assayed for telomerase activity. Given the well-established general upregulation of telomerase activity in cycling cells, it seemed likely that the relative unresponsiveness of the WT cells to CHIR was explained by their relatively diminished freedom to divide under these conditions. Indeed, upon repeating the experiment at lower cell densities, which permitted proliferation of both genotypes, we observed upregulation of telomerase activity with CHIR in both genotypes, with more dramatic upregulation in the mutant cultures but only up to the same absolute level as WT (i.e. CHIR restored telomerase levels to that of WT). We have updated the manuscript to reflect this data (Figure 4 G). This doesn’t impact the major conclusions of the study, and still indicates that upregulation of telomerase activity contributes to the rescue by CHIR99021 of mutant organoid growth.

The reviewer is correct that TERC overexpression rescues telomerase activity in *DKC1* mutant cells. But it is not correct that it has been shown that elevated TERT do not provide rescue in these mutant. Indeed, when TERT levels are limiting, artificial expression of TERT does enhance telomerase activity in *DKC1* mutant cells (Wong and Collins, 2006) (See Figure 1 of Wong and Collins 2006). While we have not directly tested if TERT levels are limiting in our cultured iAT2 cells, it is likely that they are like most other human cell types where TERT levels are limiting. With regard to *TERC,* although we were able to discern a clear decrease in *TERC* (and *TERT*) mRNA levels in the mutant iAT2 cells by ddPCR (see Figure 3—figure supplement 1), technical limitations (described above in Point 1) prevented us from establishing clear rescue of *TERC* levels under conditions of rescue by CHIR99021. However, given that *TERC* levels are generally upregulated by Myc, and that Myc is generally upregulated by GSK3 inhibition, there is a good possibility that CHIR99021 does indeed upregulate *TERC* levels, which could also contribute to the elevated telomerase activity. Regardless of the underlying mechanisms, the restoration of telomerase activity by GSK3 inhibition is of potential translational therapeutic importance.

We agree that the lack of telomere lengthening by TeSLA is surprising, as it is not what our group has seen previously with intestinal organoids with similar mutations. As we mentioned in the manuscript, we cannot rule out lengthening of the shortest telomeres that are too short to be detected by TeSLA. We also note that mean telomere lengths were actually increased by ~20%. Finally, TERT may be enhancing capping without lengthening by binding to uncapped telomeres and displacing DDR factors (Perera et al., 2019).

To address the above issues, we have modified the Results section as follows:

“We ultimately investigated how CHIR99021 affects the telomere status of DC iAT2s. CHIR treated iAT2 cells showed fewer cells with 53BP1 foci, fewer p21+ cells and fewer TIF+ cells (Figure 4 D-F). TeSLA revealed a modest increase in average telomere length but no apparent decrease in the frequency of shortest detectable telomeres (Figure 4—figure supplement 3 A-C). iAT2 cell telomerase activity was restored in mutant cells treated with CHIR99021 to levels similar to those seen in WT cells (Figure 4G). Together, these data argue that GSK3 inhibition not only rescues the growth of DC iAT2 cells, but it also rescues telomere defects, most likely through upregulation of telomerase activity that could extend telomeres that are shorter than those that can be detected by TeSLA; it is also possible that telomerase is contributing to telomere capping via lengthening-independent mechanisms (Perera et al., 2019). Of note, withdrawal of CHIR99021 during only the 14 days prior to D70 was sufficient to elicit the growth defect (Figure 4—figure supplement 1 A-B), consistent with an only modest increase in telomere length associated with the increased telomerase activity.”

Reviewer #1 (Recommendations for the authors):– The analysis shown in Figure 1C should be done for the experiment performed with iPSCs 5 passages after introduction of the DKC1 A386T mutation.

Please see our response above to Essential Revision 2. Furthermore, we have quantified organoid formation efficiency for the cultures derived from passage 5 iPSCs, as requested (see Figure S3E), and as expected, the efficiencies are indistinguishable between the wild type and mutant cultures. We were unable to accurately quantify organoid areas given the quality of the images we obtained for these particular cultures, but numerous direct observations never revealed any obvious differences in organoid sizes between wild type and mutant under these conditions.

– An apoptosis positive control is lacking for Figure S3E.

We appreciate this suggestion, and we have added a positive control to Figure 2—figure supplement 1 C and explained it in detail within the figure legend.

– The figure legend for Figure S4B does not make sense nor correlate with the main text description (lines 209-211). It appears that an experiment was designed whereby CHIR99021 was either present or removed after D56 and the cells analyzed at D70 (Figure S4A) but the data are not shown in Figure S4B.

We agree that the figure legend may not adequately reflect the data presented and the main text. We have included your suggestions and believe they help clarify the experiment performed. To simplify presentation of data, previous Figure S4 has been divided into new Figure 4—figure supplements 1, 2 and 3, and the relevant data is in the Figure 4—figure supplement 1.

– The graphics Figure 1A and S3C show Day 49 but the text reads Day 50 – this is confusing.

We appreciate how this might be confusing and have changed references to Day 49 to Day 50.

– It is not stated whether the analyses performed in Figure 2 are using cells derived from iPSCs 25 passages after introduction of the DKC1 mutation (It is likely this is the case, but it should be clearly stated).

We have changed the heading titles as well as the figure titles to reflect this more clearly.

– Line 138 – Should refer to Figure S3C-D.

We agree and have changed the reference to the supplementary figures.

– The methods section should be carefully reviewed for formatting consistency throughout – e.g.., use of capitalization or not.

We appreciate the reviewer’s suggestion and have made the requested edits to the experimental procedures section.

Reviewer #2 (Recommendations for the authors):1. The mutant cells showed down-regulation of shelterin component genes (TINF2 and POT1) downstream of the Wnt signaling pathway. To show that the GSK3 inhibitor CHIR99021 acted through Wnt signaling in iAT2 cells, the authors should show that the expression of these genes can be restored by the treatment.

See our response to essential revision 4 above.

2. Is Wnt signaling required for CHIR99021 to show its effects on cellular senescence and telomere dysfunction? This should be examined and discussed.

We are unsure of precisely what the reviewer is asking for here, but we imagine that they might be thinking of a couple of different and important issues. First, we want to address the possibility that CHIR99021 might have non-GSK3 dependent effects (i.e. how specific an inhibitor of GSK3 is CHIR99021)? To our knowledge, it is the most selective GSK3 inhibitor available (An et al., 2012), and we direct the reviewers to the original paper addressing this issue which shows it is >500-fold selective for GSKs as compared to many other kinases (Ring et al., 2003). Furthermore, the fact that we observe similar rescue with a different GSK3 inhibitor (CHIR98014) supports a role for GSK3 (and thus Wnt). Second, we want to address the possibility that the precise nature of GSK3 inhibition by CHIR99021 vs. that by Wnt might cause different effects downstream of GSK3. There are indeed multiple different possible downstream effects of the inhibition of GSK3, and it’s conceivable that these are impacted differentially by different GSK3 inhibitors. We don’t know the answer to this question, but as we describe in the manuscript, the *DKC1* mutation induces transcriptional changes in the Wnt pathway and CHIR99021 rescues the growth and telomere phenotypes. Furthermore, CHIR99021 is commonly used to replace Wnt ligands in multiple different differentiation protocols (Loh et al., 2016; Pagliuca et al., 2014; Rezania et al., 2014; Teo et al., 2014), supporting its similarity to Wnts. However, we are aware that canonical Wnts and CHIR99021 are not exactly the same in terms of their activation kinetics (Massey et al., 2019), and don’t discount the possibility that this could impact effects on downstream targets. These are important issues under active investigation, but we feel that they are outside the scope of our manuscript.

We now include these important caveats in the Discussion, as follows:

“We note that the effects of CHIR99021 could differ in some ways from canonical Wnt signaling, e.g. by impacting targets other than GSK3 or by differentially affecting the numerous pathways that are downstream of GSK3. However, the Wnt-related transcriptional changes in the mutant iAT2s, the high selectivity of CHIR99021 for GSK3 and its common use as a replacement for Wnt ligands in cell culture differentiation protocols, along with the fact that two different GSK3 inhibitors rescued growth of the mutant iAT2s suggest that the rescue provided by CHIR99021 is indeed related to GSK3 and Wnt (An et al., 2012; Loh et al., 2016; Pagliuca et al., 2014; Rezania et al., 2014; Ring et al., 2003; Teo et al., 2014).”

3. The authors showed little change in telomere length, but increased telomerase activity by CHIR99021 treatment in iAT2 cells. Some further discussion on this will be helpful for readers to understand why this is the case.

Thank you for this suggestion. Please see our response to Essential Revision 4, above.

The most relevant part of that response is reproduced here:

“We agree that the lack of telomere lengthening by TeSLA is surprising, as it is not what our group has seen previously with intestinal organoids with similar mutations. As we mentioned in the manuscript, we cannot rule out lengthening of the shortest telomeres that are too short to be detected by TeSLA. We also note that mean telomere lengths were actually increased by ~20%. Finally, TERT may be enhancing capping without lengthening by binding to uncapped telomeres and displacing DDR factors (Perera et al., 2019).”4. Wnt has been shown to regulate TERT expression. Hence, it should be tested here whether increased telomerase activity is attributed to increased TERT expression and/or TERC expression.

Again, please see our response to Essential Revision 4, above.

In particular, as the reviewer points out, it is well established that TERT is Wnt target gene in diverse contexts, and its upregulation presumably contributes to the increase in telomerase activity that we observe in CHIR-treated cultures. Nonetheless, given that *TERC* levels are generally upregulated by Myc, and that Myc is generally upregulated by GSK3 inhibition, there is a good possibility that CHIR does indeed upregulate *TERC* levels, which could also contribute to the elevated telomerase activity.

References

An, W.F., Germain, A.R., Bishop, J.A., Nag, P.P., Metkar, S., Ketterman, J., Walk, M., Weiwer, M., Liu, X., Patnaik, D., et al. (2012). Discovery of Potent and Highly Selective Inhibitors of GSK3b. In Probe Reports from the NIH Molecular Libraries Program, (Bethesda (MD): National Center for Biotechnology Information (US)),.

Cesare, A.J., Kaul, Z., Cohen, S.B., Napier, C.E., Pickett, H.A., Neumann, A.A., and Reddel, R.R. (2009). Spontaneous occurrence of telomeric DNA damage response in the absence of chromosome fusions. Nat. Struct. Mol. Biol. *16*, 1244–1251.

Cesare, A.J., Hayashi, M.T., Crabbe, L., and Karlseder, J. (2013). The telomere deprotection response is functionally distinct from the genomic DNA damage response. Mol. Cell *51*, 141–155.

Kaul, Z., Cesare, A.J., Huschtscha, L.I., Neumann, A.A., and Reddel, R.R. (2011). Five dysfunctional telomeres predict onset of senescence in human cells. EMBO Rep. *13*, 52–59.

Loh, K.M., Chen, A., Koh, P.W., Deng, T.Z., Sinha, R., Tsai, J.M., Barkal, A.A., Shen, K.Y., Jain, R., Morganti, R.M., et al. (2016). Mapping the Pairwise Choices Leading from Pluripotency to Human Bone, Heart, and Other Mesoderm Cell Types. Cell *166*, 451–467.

Martin-Ruiz, C., Saretzki, G., Petrie, J., Ladhoff, J., Jeyapalan, J., Wei, W., Sedivy, J., and von Zglinicki, T. (2004). Stochastic variation in telomere shortening rate causes heterogeneity of human fibroblast replicative life span. J. Biol. Chem. *279*, 17826–17833.

Massey, J., Liu, Y., Alvarenga, O., Saez, T., Schmerer, M., and Warmflash, A. (2019). Synergy with TGFβ ligands switches WNT pathway dynamics from transient to sustained during human pluripotent cell differentiation. Proc. Natl. Acad. Sci. U. S. A. *116*, 4989–4998.

Pagliuca, F.W., Millman, J.R., Gürtler, M., Segel, M., Van Dervort, A., Ryu, J.H., Peterson, Q.P., Greiner, D., and Melton, D.A. (2014). Generation of functional human pancreatic β cells in vitro. Cell *159*, 428–439.

Perera, O.N., Sobinoff, A.P., Teber, E.T., Harman, A., Maritz, M.F., Yang, S.F., Pickett, H.A., Cesare, A.J., Arthur, J.W., MacKenzie, K.L., et al. (2019). Telomerase promotes formation of a telomere protective complex in cancer cells. Sci Adv *5*, eaav4409.

Rezania, A., Bruin, J.E., Arora, P., Rubin, A., Batushansky, I., Asadi, A., O’Dwyer, S., Quiskamp, N., Mojibian, M., Albrecht, T., et al. (2014). Reversal of diabetes with insulin-producing cells derived in vitro from human pluripotent stem cells. Nat. Biotechnol. *32*, 1121–1133.

Ring, D.B., Johnson, K.W., Henriksen, E.J., Nuss, J.M., Goff, D., Kinnick, T.R., Ma, S.T., Reeder, J.W., Samuels, I., Slabiak, T., et al. (2003). Selective glycogen synthase kinase 3 inhibitors potentiate insulin activation of glucose transport and utilization in vitro and in vivo. Diabetes *52*, 588–595.

Ruis, P., and Boulton, S.J. (2021a). The end protection problem—an unexpected twist in the tail. Genes & Development *35*, 1–21.

Ruis, P., and Boulton, S.J. (2021b). The end protection problem—an unexpected twist in the tail. Genes & Development *35*, 1–21.

Suram, A., and Herbig, U. (2014). The replicometer is broken: telomeres activate cellular senescence in response to genotoxic stresses. Aging Cell *13*, 780–786.

Teo, A.K.K., Valdez, I.A., Dirice, E., and Kulkarni, R.N. (2014). Comparable generation of activin-induced definitive endoderm via additive Wnt or BMP signaling in absence of serum. Stem Cell Reports *3*, 5–14.

Van Ly, D., Low, R.R.J., Frölich, S., Bartolec, T.K., Kafer, G.R., Pickett, H.A., Gaus, K., and Cesare, A.J. (2018a). Telomere Loop Dynamics in Chromosome End Protection. Mol. Cell *71*, 510–525.e6.

Van Ly, D., Low, R.R.J., Frölich, S., Bartolec, T.K., Kafer, G.R., Pickett, H.A., Gaus, K., and Cesare, A.J. (2018b). Telomere Loop Dynamics in Chromosome End Protection. Mol. Cell *71*, 510–525.e6.

Wong, J.M.Y., and Collins, K. (2006). Telomerase RNA level limits telomere maintenance in X-linked dyskeratosis congenita. Genes Dev. *20*, 2848–2858.